

# A turbulence data reduction scheme for autonomous and expendable profiling floats

Kenneth G. Hughes[1], James N. Moum[1], and Daniel L. Rudnick[2]

[1]College of Earth, Ocean, and Atmospheric Sciences, Oregon State University, Corvallis, Oregon.
[2]Scripps Institution of Oceanography, University of California, San Diego, La Jolla, California.

**Correspondence:** Kenneth Hughes (kenneth.hughes@oregonstate.edu)

**Abstract.** Autonomous and expendable profiling float arrays such as deployed in the Argo Program require the transmission of reliable data from remote sites. However, existing satellite data transfer rates preclude complete transmission of rapidly sampled turbulence measurements. It is therefore necessary to reduce turbulence data onboard. Here we propose a scheme for onboard data reduction and test it with existing turbulence data obtained with a newly developed version of a SOLO-II profiling float. The scheme invokes simple power law fits to (i) shear probe voltage spectra and (ii) fast thermistor voltage spectra that yield a fit value plus a quality control metric. At roughly 1 m vertical interval resolution, this scheme reduces the necessary data transfer volume 240-fold to approximately 3 kB for every 100 m of a profile (when profiling at $0.2 \, \mathrm{m \, s^{-1}}$). Turbulent kinetic energy dissipation rate $\varepsilon$ and thermal variance dissipation rate $\chi$ are recovered in post-processing. As a test, we apply our scheme to a dataset comprising 650 profiles and compare its output to that from our standard turbulence processing algorithm. For $\varepsilon$, values from the two approaches agree within a factor of two 87% of the time; for $\chi$, 78%. These levels of agreement are greater than or comparable to that between the $\varepsilon$ and $\chi$ values derived from two shear probes and two fast thermistors, respectively, on the same profiler.

## 1 Introduction

Measurements of oceanic turbulence have been made since the 1950s using platforms and sensors of various shapes and sizes (Lueck et al. 2002). Complete resolution of the turbulence requires measuring temperature and velocity gradients at millimeter-to-centimeter scale. Hence, sampling turbulence is data intensive. Whereas conventional profiling measurements of temperature, conductivity, and pressure are





typically sampled at 1 Hz (e.g., Argo floats; Roemmich et al. 2019a), a turbulence profile involves sampling multiple sensors at 100–1000 Hz. A relatively minimal requirement of five separate signals sampled at 100 Hz and recorded at 16-bit resolution equates to $1 \, \text{kB s}^{-1}$, or 500 kB per 100 m of profiling range at $0.2 \, \text{m s}^{-1}$ profiling speed. For floats, this is not a trivial volume of data. For example, transmitting only 3 kB of data from a Deep SOLO float takes 100–200 s (Roemmich et al. 2019b). Extended surfacings also

present a danger from surface vessels and vandals. Ultimately, raw turbulence profiles are two-to-three orders of magnitude too large to transmit in a reasonable amount of time.

One approach to reducing turbulence data is given by Rainville et al. (2017) who use it for multi-month glider missions. Onboard the glider, they calculate spectra of raw voltage signals reported by the shear probes and fast thermistors and then band average each of these spectra into 12 bins. After transmis-

sion, these binned values are calibrated and fit to model spectra. Although we will share this strategy of postponing calibration, our scheme differs from Rainville et al. (2017). Instead, we more closely follow Becherer and Moum (2017) who designed a scheme to reduce moored $\chi$pod data by more than four orders of magnitude. Overall, our goal is to minimize the file size to be transmitted, and yet also minimize the amount by which we manipulate and process the data onboard.

Becherer and Moum (2017) showed that, for a given segment, turbulence quantities can be reconstituted from voltage quantities (means, variances, and power law fits). We adapt their approach so that it works for a vertical profiler (Sect. 2). First, we document the necessary calibration details (Sect. 3). Next, we compress raw shear voltages by way of simple power law fits (Sect. 4). A test of the scheme employing 650 profiles demonstrates that little accuracy is sacrificed in return for a large reduction in data volume

(Sect. 5). A similar method and test is given for fast thermistor measurements (Sect. 6 and 7). Adapting the scheme to a different profiler requires minimal modification (Sect. 8). For our particular profiler, the scheme reduces the dataset size by a factor of $\sim$240: only 3 kB for each 100 m of a profile (Sect. 9).

## 2   The Flippin' $\chi$SOLO (FCS)

We intend our data reduction scheme to be sufficiently general to be portable to all vertical turbulence

profilers. It can also be used with gliders if a measure of flow speed past the sensors is available (e.g., Greenan et al. 2001; Merckelbach and Carpenter 2021). In a general sense, some of the values specified





herein ought to be considered variables (Sect. 8). However, we do have a particular platform for which we are developing the scheme: the Flippin' $\chi$SOLO (FCS), and the values used here are chosen for the objective of detailed upper ocean profiling.

FCS is a conventional SOLO-II profiling float (Roemmich et al. 2004) with the addition of a turbulence package plus extra functionality. The turbulence package includes two shear probes (Osborn 1974) to measure small-scale velocity gradients from which $\varepsilon$ is computed, two fast thermistors to measure small-scale temperature fluctuations from which $\chi$ is computed, as well as a pitot tube (Moum 2015), pressure sensor, three-axis accelerometer, and compass. The pressure sensor yields a measure of profiling speed

used in our scheme. The pitot tube, accelerometer, and compass data are not used in the turbulence data reduction scheme but have other purposes such as measuring the surface wave field. When changing its buoyancy to switch profiling direction, FCS also flips (via internal shifting of the battery pack) so that the turbulence sensors always point into undisturbed fluid. Flipping therefore permits profiling on both ascent and descent including sampling of the upper 5 m on the upward profile. FCS and its measurements will

be described more completely in a future paper.

As a prototype, a SOLO float without flipping capability but with a modified $\chi$pod (Moum and Nash 2009) attached was deployed in the Bay of Bengal to measure the suppression of turbulence by salinity stratification (Shroyer et al. 2016). Two units with flipping capabilities and fully integrated turbulence packages were subsequently built and vetted over four days in May 2019 off the Oregon coast. During

this period, each unit profiled from the surface to $\sim$120 m and back at a typical speed of 0.2 m s$^{-1}$. Adding time at the surface, each dive cycle took $\sim$30 minutes and we obtained 650 profiles in total.

In this 2019 experiment, one of the shear probes on one of the two units malfunctioned. Hence, the dataset for this paper contains approximately 25% fewer shear data than fast thermistor data.

## 3    Conversion of measured voltages to physical units

The core of our data reduction scheme uses power law fits of voltage spectra that are calculated onboard, and subsequently converted to meaningful turbulence quantities in post-processing. Additional voltage quantities are also recorded to determine temperature, pressure, and profiling speed.



## 3.1  Nomenclature and conventions

- All quantities measured by FCS that are discussed in this paper are sampled at 100 Hz.

- All voltage spectra are frequency spectra and denoted $\Psi_x(f)$ (where $x$ is a label such as $s$ for shear) with units of $\mathrm{V^2\,Hz^{-1}}$.

- Physical spectra of shear and temperature gradient are wavenumber spectra and denoted $\Phi_x(k)$ with units of $\mathrm{s^{-2}\,cpm^{-1}}$ and $\mathrm{K^2\,m^{-2}\,cpm^{-1}}$, respectively. Figure 4 is an exception in which shear spectra are frequency spectra: $\Phi_s(f)$.

- Wavenumber $k$ has the unit cycles per meter (cpm). Expressions quoted from other papers may differ by factors of $2\pi$ for wavenumbers in radians per meter.

- The Kraichnan model spectrum $\Phi_{\mathrm{Kr}}$ primarily depends on the dissipation rates of turbulent kinetic energy and temperature variance ($\varepsilon$ and $\chi$), but it also depends on the molecular viscosity $\nu$ and molecular thermal diffusivity $D_T$. For brevity, we write $\Phi_{\mathrm{Kr}}(k,\varepsilon,\chi)$ rather than the more complete $\Phi_{\mathrm{Kr}}(k,\varepsilon,\chi,\nu,D_T)$. Similarly, the Nasmyth spectrum is written $\Phi_{\mathrm{Na}}(k,\varepsilon)$ rather than $\Phi_{\mathrm{Na}}(k,\varepsilon,\nu)$. In cases where the arguments are unambiguous or unimportant, we simply write $\Phi_{\mathrm{Na}}$ and $\Phi_{\mathrm{Kr}}$.

- A pair of angle brackets, $\langle\cdot\rangle$, denotes the mean value over a segment of length $N_{\mathrm{seg}} = 512$ data points. This equates to $\sim 1\,\mathrm{m}$ at our nominal profiling speed of $0.2\,\mathrm{m\,s^{-1}}$.

- To calculate spectra for a given 512-element voltage segment, we first remove the linear trend, then use three half-overlapping, Hamming-windowed, 256-element subsegments (i.e., $N_{\mathrm{fft}} = 256, N_{\mathrm{overlap}} = 128$).

In general, the values of $N_{\mathrm{seg}}$ and $N_{\mathrm{fft}}$ are variables. Our choices are based on the 100 Hz sampling ($\sim 500\,\mathrm{cpm}$) and the goals of FCS, which include obtaining high-vertical-resolution turbulence data, especially near the surface. For different turbulence profilers or different scientific goals, longer segments and/or more overlapping subsegments may be more appropriate (see Sect. 8).



## 3.2 Shear calibration

The voltage reported by the shear probe $V_s$ is linearly proportional to shear:

$$u_z = \frac{\alpha}{W^2} V_s \tag{1}$$

$$\alpha = 1/(2\sqrt{2}\rho G_s T_s S_s) \tag{2}$$

where $W$ is the flow speed past the sensor. The overall engineering calibration $\alpha$ includes the seawater density $\rho$, the analog circuit gain $G_s$ (equal to 1 for FCS circuitry), the probe sensitivity $S_s$ ($\sim 0.25 \times 10^{-3}\,\mathrm{V\,m^2\,N^{-1}}$) and the differentiator time constant $T_s$ ($\sim 1\,\mathrm{s}$).

The linearity in $V_s$ admits a simple link between the physical and voltage spectra:

$$\Phi_{u_z}(k) = \frac{1}{H_s^2(k)} \frac{\alpha^2}{W^3} \Psi_s(f). \tag{3}$$

where $H_s^2(k)$ is the transfer function that accounts for (i) spatial averaging by the shear probe of high-wavenumber motions and (ii) analog and digital filtering of the raw voltage signal (see Appendix A). Note also the use above of the following relation:

$$\Psi(f) = \Psi(k)\frac{\mathrm{d}k}{\mathrm{d}f} = \frac{\Psi(k)}{W}. \tag{4}$$

### 3.3 Temperature and temperature gradient calibration

Two voltage signals are recorded for each fast thermistor. $V_T$ is the voltage output directly related to $T$ and $V_{Tt}$ is the differentiated output, which improves resolution at high frequencies ($\gtrsim 10\,\mathrm{Hz}$). Temperature is related to $V_T$ through a quadratic calibration:

$$T = C_{1T} + C_{2T}V_T + C_{3T}V_T^2 \tag{5}$$

$$\langle T \rangle = C_{1T} + C_{2T}\langle V_T \rangle + C_{3T}\langle V_T^2 \rangle \tag{6}$$

where $C_{1T}$, $C_{2T}$, and $C_{3T}$ are coefficients determined from lab calibrations. Note how $\langle T \rangle$ depends on the means of both $V_T$ and $V_T^2$ because of the quadratic calibration.

The gradient of this calibration is

$$\frac{\partial T}{\partial V_T} = C_{2T} + 2C_{3T}V_T \approx C_{2T} + 2C_{3T}\langle V_T \rangle. \tag{7}$$





Over 5 s time scales, we consider $V_T$ to be constant. Consequently, the small-scale vertical temperature gradient $T_z$ is linearly proportional to the differentiated voltage $V_{Tt}$. To demonstrate, we first rewrite $T_z$ in terms of more directly measured quantities:

$$T_z = \frac{\partial T}{\partial z} = \frac{\partial T}{\partial V_T} \frac{\partial V_T}{\partial t} \frac{\partial t}{\partial z}. \tag{8}$$

The first quantity on the right-hand side is Eq. (7), the last is $1/W$, and the second is

$$\frac{\partial V_T}{\partial t} = \frac{V_{Tt}}{C_{Tt}} \tag{9}$$

where $C_{Tt}$ is the gain of the analog differentiator.

Rewriting Eq. (8), the aforementioned linear relationship between $T_z$ and $V_{Tt}$ becomes

$$T_z = \left( \frac{C_{2T} + 2C_{3T} \langle V_T \rangle}{C_{Tt} W} \right) V_{Tt}. \tag{10}$$

The relationship between physical and voltage spectra is therefore

$$\Phi_{T_z}(k) = \left( \frac{C_{2T} + 2C_{3T} \langle V_T \rangle}{C_{Tt}} \right)^2 \frac{1}{W} \Psi_{Tt}(f). \tag{11}$$

Again, we have invoked Eq. (4).

### 3.4 Pressure and profiling velocity calibration

Pressure has a linear calibration:

$$P\,[\text{dbar}] = \frac{C_{1P} + C_{2P} V_P}{1.45\,\text{psi}\,\text{dbar}^{-1}} - p_{\text{atm}}. \tag{12}$$

In our usage, the coefficients $C_{1P}$ and $C_{2P}$ are recorded in units of psi and psi V$^{-1}$, respectively, and calibrated under total pressure. Subtracting atmospheric pressure makes $P = 0$ at the sea surface. The constant $C_{1P}$ must account for the vertical position of the pressure sensor on the instrument relative to the shear probes and thermistors. Hence, $C_{1P}$ differs between upcasts and downcasts. For the reduced dataset, we want $\langle P \rangle$, which is simply Eq. (12) with $\langle V_P \rangle$ in place of $V_P$.

The flow speed past the sensors, denoted $W$, is derived from the pressure voltage rate of change. Over a segment of length $N$, the mean of $W$ is a scaled version of the difference between the first and $N$th voltage



values:

$$\langle W \rangle = \left\langle \frac{\partial P}{\partial t} \right\rangle = \frac{C_{2P}}{1.45} \underbrace{\frac{|\Delta V_P|}{(N-1)\Delta t}}_{\text{Calculate onboard}} \tag{13}$$

where $\Delta t$ is the sampling period (here 0.01 s), and $\Delta V_P$ is $V_P(N) - V_P(1)$. No smoothing is necessary before calculating $\Delta V_P$ because its magnitude is so much larger than the quantization of the signal (this being the limiting factor for precision of pressure recorded by FCS). In physical units, $P$ is precise to 0.003 dbar, which is $\mathcal{O}(300)$ times smaller than $\Delta P$.

Wave orbitals can introduce variability when $W$ is small ($\lesssim 0.15\,\mathrm{m\,s^{-1}}$). As a diagnostic we calculate and record the minimum value of $W$ for each segment. This also helps to identify the beginning and end of profiles as shown in Appendix B. In standard processing, we would derive $W(t)$ from the pressure signal low passed at 2 Hz. To avoid the need to low-pass filter the signal onboard, we instead make 10 estimates of $W(t)$ per segment and take the minimum of these:

$$W_{\min} = \frac{C_{2P}}{1.45} \underbrace{\min\left( \frac{|\Delta V_P(t_i)|}{50\Delta t} \right)}_{\text{Calculate onboard}} \tag{14}$$

where $t_i = 1, 51, 101, \ldots, 501$. Even with this sampling of every 50th element, which follows from subsampling a 100 Hz signal at 2 Hz, $\Delta V_P(t_i)$ is large enough that smoothing is unnecessary.

In this paper, we immediately discard all segments in which $W_{\min} < 0.05\,\mathrm{m\,s^{-1}}$. This threshold is reached only at the top and bottom of profiles, if at all. Note, however, that this does not imply that a segment with $W_{\min} > 0.05\,\mathrm{m\,s^{-1}}$ is trustworthy. Even segments with $W_{\min}$ closer to $0.15\,\mathrm{m\,s^{-1}}$ should be treated with particular caution. Signs that a segment is questionable are that $\langle W \rangle$ and $W_{\min}$ differ by more than $\sim 20\%$ or that spectral fit scores are low (see Sect. 4.3 and 6.3).

# 4 Reduction of shear data

## 4.1 Summarizing Nasmyth spectra with $f^{1/3}$ fits

Shear measurements ideally capture both the inertial and viscous subranges and hence use a wide band of the measured spectrum to derive values for $\varepsilon$. In practice, noise and sensor resolution limit how well the



true environmental spectrum is resolved. Conventional work-arounds exploit the Nasmyth model spec-
trum $\Phi_{\mathrm{Na}}(k,\varepsilon)$ (Nasmyth 1970; Oakey 1982). One approach is to iterate toward a solution in which the
integral of $\Phi_{\mathrm{Na}}$ over a specific wavenumber band matches that of the measured spectrum $\Phi_{u_z}$ (e.g., Moum
et al. 1995). Another is to find the best fit of $\Phi_{u_z}$ to $\Phi_{\mathrm{Na}}$ by using maximum likelihood estimation together
with a model of the expected statistical distribution of the spectral coefficients being fitted (e.g., Bluteau
et al. 2016).

Here we develop a new and simpler two-stage approach to fitting shear spectra to $\Phi_{\mathrm{Na}}$. In the first stage,
we use an $f^{1/3}$ power law fit over a fixed frequency range of $f_l$ to $f_h = 1–5$ Hz, where $f^{1/3}$ follows from the
assumption that we are fitting over the inertial subrange. In this subrange, shear spectra are proportional
to $k^{1/3}$ and hence also $f^{1/3}$ since $f = Wk$. With $N_{\mathrm{fft}} = 256$ and $100$ Hz sampling (Sect. 3.1), spectral
coefficients are separated by frequency increments of $100$ Hz/$256 = 0.39$ Hz, so there are 10 coefficients
between 1 and 5 Hz. (Our processing code will actually use bounding frequencies of 0.98 and 4.88 Hz as
these are half-integer multiples of 0.39 Hz, but for brevity we will write these as 1 and 5 Hz throughout.)

Our choice of $f_l = 1$ Hz is dictated by a requirement that we avoid low frequency contamination induced
by (i) advection by wave orbital motion and (ii) pitch and roll motions of the profiler. Together, these
dominate below 0.3 Hz. Setting $f_l = 0.5$ Hz would add only one more spectral coefficient. Our choice of
$f_h = 5$ Hz is a trade-off between maximizing the bandwidth of the fit and minimizing how much measured
spectra are subject to either noise or viscous roll off. Other profilers may benefit from different frequency
bounds (see Sect. 8).

Our inertial subrange assumption is often false. Indeed, 'assumption' is perhaps a misnomer as we
do not expect it to be true; we know that viscous roll off will often occur at frequencies lower than 5 Hz
(25 cpm for a nominal value of $W = 0.2\,\mathrm{m\,s^{-1}}$). However, because there exists an analytical expression for
the viscous roll off, we are able to derive an exact expression that quantifies how much $\varepsilon$ is underestimated.
This is the second stage of our approach. We derive an expression for the correction function $F_{\mathrm{Na}}$ in such
a way that it can be calculated in post-processing. The benefits of this approach are that (i) we can fit
uncalibrated (i.e., voltage) spectra and (ii) it simplifies the actual onboard fitting routine (Sect. 4.2).





The full Nasmyth spectra and its inertial range approximation are as follows (Lueck 2013):

$$\Phi_{Na}(k,\varepsilon) = \frac{\varepsilon^{3/4}}{\nu^{1/4}} \frac{8.05(k\eta)^{1/3}}{1+(20.6k\eta)^{3.715}} \tag{15}$$

$$\Phi_{Na}(k \lesssim 0.02/\eta, \varepsilon) = 8.05 k^{1/3} \varepsilon^{2/3} \tag{16}$$

where $\eta = (\nu^3/\varepsilon)^{1/4}$ is the Kolmogorov length scale.

Consider an $f^{1/3}$ fit of the Nasmyth spectrum over $f_l$–$f_h = 1$–5 Hz for two values of $\varepsilon$: $1 \times 10^{-9}$ and

$1 \times 10^{-6}$ W kg$^{-1}$ (Fig. 1a). With our nominal value of $W = 0.2$ m s$^{-1}$, we get $k_l$–$k_h = 5$–25 cpm. For $\varepsilon = 10^{-6}$ W kg$^{-1}$ the $f^{1/3}$ fit lies on top of $\Phi_{Na}$. Conversely, the $f^{1/3}$ fit for the smaller $\varepsilon$ value is seemingly meaningless: the $f^{1/3}$ fit (dashed line) does not even match the sign of the slope of $\Phi_{Na}$. Worse yet, naively inverting this *initial* (or 'init') fit produces the underestimate $\varepsilon_{init} = 1.2 \times 10^{-10}$ W kg$^{-1}$, six times smaller than the true value of $\varepsilon$. However, by adjusting by a factor of $1/F_{Na}$, defined in the following paragraph,

the fit (dotted line) now looks like a hypothetical extrapolation of the inertial subrange. Equivalently, $\varepsilon_{init}$ is corrected to the true value of $\varepsilon$ as

$$\varepsilon = \varepsilon_{init}/F_{Na}^{3/2}. \tag{17}$$

In our example, $1 \times 10^{-9}$ W kg$^{-1}$ = $1.2 \times 10^{-10}$ W kg$^{-1}/0.238^{3/2}$. The value of 0.238 is the solution to an implicit equation derived below that depends on $\varepsilon_{init}$ and $W$. For clarity, our demonstration starts by

assuming we know $\varepsilon$ rather than $\varepsilon_{init}$.

Nasmyth spectra can be flattened to unity over the inertial subrange with the normalization $8.05 k^{1/3} \varepsilon^{2/3}$ (Fig. 1b). Values of $F_{Na}$ are based on the mean of these flattened spectra over the wavenumber range $k_l$–$k_h$ ($= f_l/W$–$f_h/W$):

$$F_{Na} = \frac{1}{k_h - k_l} \int_{k_l}^{k_h} \frac{\Phi_{Na}(k,\varepsilon)}{8.05 k^{1/3} \varepsilon^{2/3}} dk. \tag{18}$$

To remove the dependence of the true value of $\varepsilon$, we substitute using Eq. (17). Further, to account for the $H_s^2(k)$ factor in Eq. (3), we make the substitution

$$\Phi_{Na}(k,\varepsilon) \rightarrow H_s^2(k) \Phi_{Na}(k,\varepsilon). \tag{19}$$

Think of this substitution as inverting the conventional way that $H_s^2(k)$ is invoked. Usually, a measured shear spectrum is amplified at high wavenumbers by $1/H_s^2(k)$ and then fit to the model spectrum $\Phi_{Na}$.



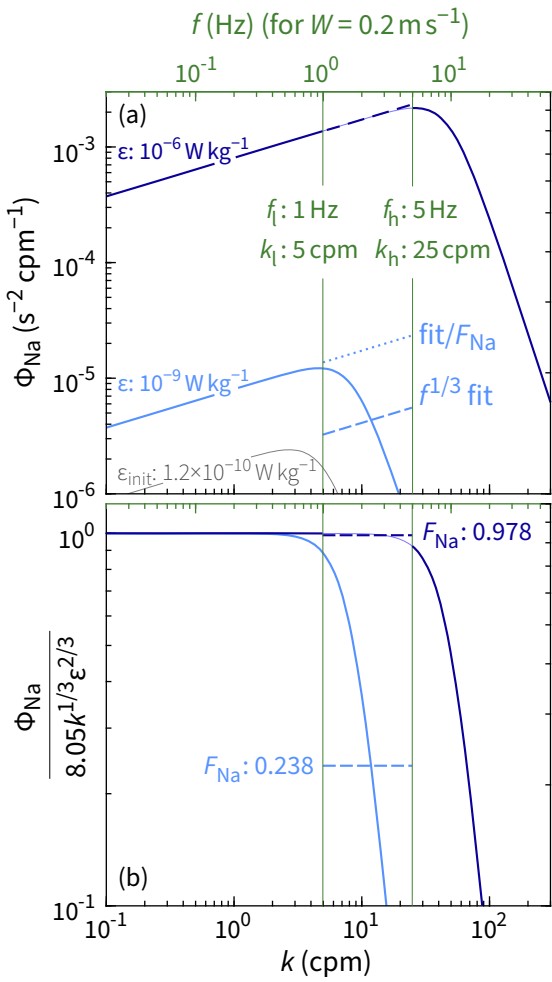

**Figure 1.** Calculation of the correction function $F_{Na}$ for two values of $\varepsilon$. For $\varepsilon = 1 \times 10^{-6}\,\mathrm{W\,kg^{-1}}$, a $k^{+1/3}$ power law is a good approximation of the Nasmyth spectrum over the frequency range $f_l$–$f_h$ (1–5 Hz) for a profiling speed of $W = 0.2\,\mathrm{m\,s^{-1}}$. Although the same is not true for $\varepsilon = 1 \times 10^{-9}\,\mathrm{W\,kg^{-1}}$, we can account for the roll off with a factor of $F_{Na}$. $F_{Na}$ can be defined in terms of either $\varepsilon$ (Eq. (18)) or $\varepsilon_{init}$ (Eq. (20)). Panel b takes the former approach. In practice, we must take the latter approach since we do not know $\varepsilon$ until after it is derived from $\varepsilon_{init}$ and $F_{Na}$.

Here, instead of amplifying the measured spectrum, we reduce the model spectrum. With this latter approach, $H_s^2(k)$ is calculated and applied only during the post-processing stage. (It changes $F_{Na}$ by only $\sim 5\%$ since we fit over relatively low wavenumbers.) Altogether, the substitutions result in an implicit





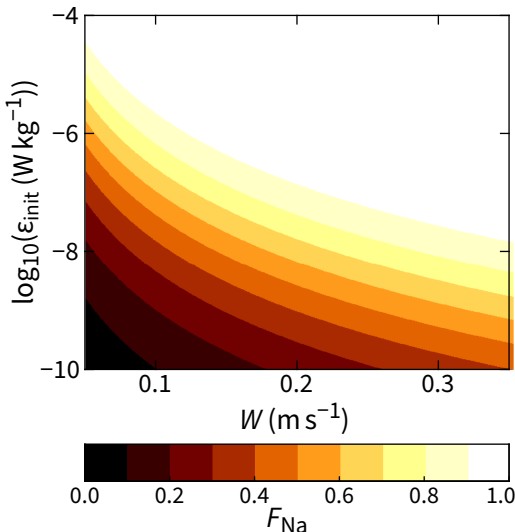

**Figure 2.** The correction function $F_{\mathrm{Na}}$ for $\nu = 1 \times 10^{-6}\,\mathrm{m^2\,s^{-1}}$ and $f_{\mathrm{l}}$–$f_{\mathrm{h}} = 1$–$5\,\mathrm{Hz}$.

function for $F_{\mathrm{Na}}$, which can be solved numerically:

$$\frac{1}{k_{\mathrm{h}} - k_{\mathrm{l}}} \int\limits_{k_{\mathrm{l}}}^{k_{\mathrm{h}}} \frac{H_s^2(k)\,\Phi_{\mathrm{Na}}(k, \varepsilon_{\mathrm{init}}/F_{\mathrm{Na}}^{3/2})}{8.05 k^{1/3} \varepsilon_{\mathrm{init}}^{2/3}/F_{\mathrm{Na}}} \mathrm{d}k - F_{\mathrm{Na}} = 0. \tag{20}$$

Note how the two forms of $F_{\mathrm{Na}}$ (Eqs. 18 and 20) are defined with different arguments. For our example, $F_{\mathrm{Na}}(\varepsilon = 1 \times 10^{-9}\,\mathrm{W\,kg^{-1}}) = F_{\mathrm{Na}}(\varepsilon_{\mathrm{init}} = 1.2 \times 10^{-10}\,\mathrm{W\,kg^{-1}}) = 0.238$. Hereafter, we use the latter: $F_{\mathrm{Na}}(\varepsilon_{\mathrm{init}})$.

With $f_{\mathrm{l}}$ and $f_{\mathrm{h}}$ fixed, $F_{\mathrm{Na}}$ is a function of three variables: $\varepsilon_{\mathrm{init}}$, $W$, and $\nu$. $F_{\mathrm{Na}}$ is closer to one (less of a correction) for larger values of $\varepsilon_{\mathrm{init}}$ (Fig. 1). It is also closer to one for higher values of $W$ (Fig. 2) since $k_{\mathrm{l}}$

and $k_{\mathrm{h}}$ decrease with increasing $W$ (i.e., $k_{\mathrm{l}}$–$k_{\mathrm{h}}$ moves closer to the inertial subrange).

## 4.2 Obtaining $\varepsilon_{\mathrm{init}}$ from shear voltage spectra

Since $\varepsilon$ can be reconstructed from $\varepsilon_{\mathrm{init}}$, we require an expression linking $\varepsilon_{\mathrm{init}}$ to the shear voltage spectrum $\Psi_s$. Equating Eq. (3) and Eq. (16) gives

$$\frac{\alpha^2}{W^3} \Psi_s(f) = 8.05 k^{1/3} \varepsilon_{\mathrm{init}}^{2/3} \tag{21}$$





where we have left out $H_s^2(k)$ since it has been incorporated into $F_{Na}$. Rearranging and substituting $k = f/W$ gives

$$\varepsilon_{init}^{2/3} f^{1/3} = \frac{\alpha^2}{8.05 W^{8/3}} \Psi_s(f).$$ (22)

Then, to solve for $\varepsilon_{init}$, we use a least-squares fit (see Appendix C):

$$\varepsilon_{init}^{2/3} = \frac{\alpha^2}{8.05 W^{8/3}} \underbrace{\frac{\sum_f f^{1/3} \Psi_s}{\sum_f f^{2/3}}}_{\text{Calculate onboard}}$$ (23)

where the sums are understood to be over the range $f_l$–$f_h$. The quantities $\alpha$, $W$, and $\varepsilon_{init}$ are calculated in post-processing.

### 4.3 Quality control of the shear spectral fits

Measured shear spectra are often quality controlled either by manual visual inspection or, more objectively, by quantifying the level of mismatch between them and their associated model. Possible mismatch

quantities include the mean absolute deviation or the variance of the ratio $\Phi_{u_z}/\Phi_{Na}$ (e.g., Ruddick et al. 2000; Bluteau et al. 2016). We cannot calculate such quantities with our reduced scheme because we do not know what each spectrum should look like until we calculate its $\varepsilon$ value in the post-processing stage. (Recall that $\Phi_{Na}$ is a function of $\varepsilon$.) By this stage, we have lost information about the spectral shape through the summing operation in Eq. (23).

To retain at least some information about the shape of each voltage spectrum, we will split the 1–5 Hz range and compute two fits rather than one. Doing so allows for a first-order check that the spectrum over the 1–5 Hz range approximately follows the expected shape.

Mathematically, there is nothing special about our choice $f_l$–$f_h$ = 1–5 Hz. In theory, we can split the 1–5 Hz range into two (1–3 Hz and 3–5 Hz) and obtain a value of $\varepsilon_{init}$ for each. These values will differ,

but so will the associated values of $F_{Na}$. For a measured spectrum that conforms to a Nasmyth spectrum, the two values of $\varepsilon$ calculated with Eq. (17) will not differ (Fig. 3). We therefore calculate onboard the sums in Eq. (23) over both $f_l$–$f_m$ and $f_m$–$f_h$, where the mid frequency $f_m$ = 3 Hz. (In our code, $f_m$ is actually $7.5 \times 0.39$ Hz = 2.93 Hz for the reason given in Sect. 4.1.) Hence, for each spectrum we are able to post-process to recover two independent estimates of $\varepsilon$, denoted $\varepsilon_{l-m}$ and $\varepsilon_{m-h}$. The mean of these





two provides a single, final value for $\varepsilon$, and their ratio quantifies the match of a measured spectrum to a
Nasmyth spectrum over the range $f_l$–$f_h$:

$$\varepsilon = \mathrm{mean}\left(\varepsilon_{l-m}, \varepsilon_{m-h}\right) \tag{24}$$

$$\varepsilon \text{ fit score} = \frac{\min\left(\varepsilon_{l-m}, \varepsilon_{m-h}\right)}{\max\left(\varepsilon_{l-m}, \varepsilon_{m-h}\right)}. \tag{25}$$

The best possible fit score is 1; the lower the score, the poorer the fit (Fig. 4). In practice, we expect a
range of $\varepsilon$ fit scores: instantaneous and unaveraged spectra differ from the Nasmyth spectrum because
they are derived from a limited sampling of a statistical process, and they can also deviate because of
non-stationarity, anisotropy, and inhomogeneity of the turbulence.

When $\varepsilon$ is small ($\lesssim 10^{-9}\,\mathrm{W\,kg^{-1}}$), the fit score may be consistently low if spectral coefficients in the
$f_m$–$f_h$ range are affected by noise and consequently $\varepsilon_{m-h} \gg \varepsilon_{l-m}$. For such cases, we choose to use only
the lower-frequency fit. We would rather have a more accurate estimate of $\varepsilon$ and forgo the fit score than
have a biased-high $\varepsilon$ value with a biased-low fit score. (Either way, the small values of $\varepsilon$ in question will
have minimal effect on any averages given that turbulence is approximately lognormal.) Specifically,

$$\left.\begin{array}{rcl} \varepsilon &=& \varepsilon_{l-m} \\ \varepsilon \text{ fit score} &=& - \end{array}\right\} \text{ if } 0.1W\left(\frac{\varepsilon_{l-m}}{v^3}\right)^{1/4} < f_m, \tag{26}$$

where the threshold is equivalent to $k < 0.1/\eta$ with $\eta$ the Kolmogorov length scale estimated from $\varepsilon_{l-m}$.
For reference, $\Phi_{Na}$ peaks at $k = 0.026/\eta$ and rolls off to 11% of its maximum by $k = 0.1/\eta$ (see Eq. (15)).

## 5   Test of the reduction scheme for $\varepsilon$

To test the accuracy of the shear reduction scheme described in the previous section, we apply it retro-
spectively to the dataset from the 2019 test cruise (Sect. 2). We compare the results to those obtained
with the standard processing scheme. This standard scheme (Appendix D) features a more sophisticated
despiking routine than used for our reduced scheme, which employs a three standard deviation threshold
filter (Appendix E).

A profile-by-profile comparison of the two schemes is shown in Fig. 5. The comparison is then extended
to all 650 profiles ($>77\,000$ segments of shear), where we find that $\varepsilon$ from the reduced scheme ($\varepsilon_{init}/F_{Na}^{3/2}$)





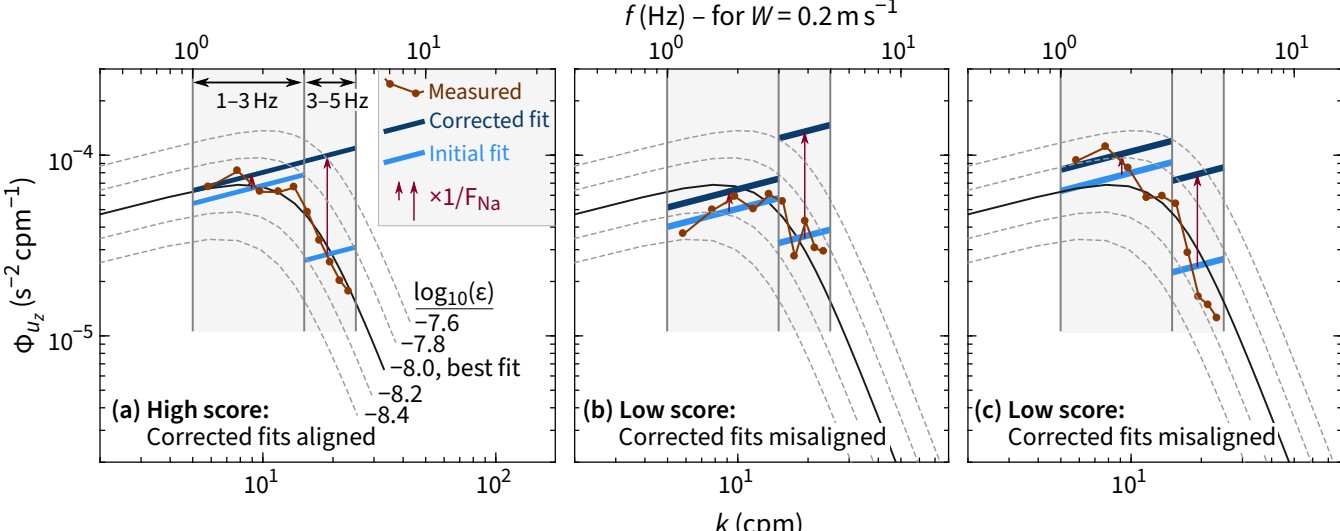

**Figure 3.** A visual demonstration of how the $\varepsilon$ fit score (Eq. (25)) characterizes better and worse fits. For all three examples, $\varepsilon$ values from the two fits (1–3 and 3–5 Hz) average to $\varepsilon = 1 \times 10^{-8}\,\mathrm{W\,kg^{-1}}$. Only in panel a, however, does the measured spectrum agree well with the Nasmyth spectrum for this $\varepsilon$ value. In practice, the initial fits would be undertaken on voltage spectra. Here, we are using physical units for simplicity.

is within a factor of two of that from the standard scheme 87% of the time over the full range of measured

values, $10^{-10} < \varepsilon < 10^{-4}\,\mathrm{W\,kg^{-1}}$ (Fig. 6a–b). For comparison, in only 72% do we obtain a factor-of-two agreement between the two independent values of $\varepsilon$ measured on the unit with two working shear probes (not shown). Further, to obtain this 87% agreement, we clearly need the correction function $F_{\mathrm{Na}}$: Fig. 6c shows that the uncorrected values $\varepsilon_{\mathrm{init}}$ only have 1:1 agreement with $\varepsilon$ from the standard scheme if $\varepsilon \gtrsim 10^{-7}\,\mathrm{W\,kg^{-1}}$. For the lowest values of $\varepsilon$, the ratio is closer to 1:30.

To demonstrate the ability of the $\varepsilon$ fit score to characterize spectra, we show two-dimensional histograms of non-dimensionalized spectral coefficients from all 77 000 measured shear spectra separated into three classes based on their $\varepsilon$ fit score: 0.67–1.00, 0.33–0.67, and 0.00–0.33. Only the lowest scoring class fails to collapse to the Nasmyth spectrum (Fig. 7c).




# 6 Reduction of fast thermistor data

The scheme to reduce fast thermistor data to enable measurement of $\chi$ is much like the scheme to reduce shear data. As in Sect. 4, we first show how we summarize a model spectrum in terms of a power law fit and a correction factor. Then we derive the implementation in terms of voltages and calculate a spectral fit metric.

## 6.1 Summarizing Kraichnan spectra with $f^1$ fits

Here we take the Kraichnan spectrum $\Phi_{\mathrm{Kr}}$ (Kraichnan 1968) as our model and for its low-wavenumber approximation we use the viscous–convective subrange, which scales as $k^{+1}$. In units of $\mathrm{K}^2\,\mathrm{m}^{-2}\,\mathrm{cpm}^{-1}$, $\Phi_{\mathrm{Kr}}$ and its approximation are as follows (e.g., Peterson and Fer 2014):

$$\Phi_{\mathrm{Kr}}(k,\varepsilon,\chi) = 4\pi^2 k\chi q\sqrt{\nu/\varepsilon}\,\exp\left(-\sqrt{6q}2\pi k\lambda_B\right) \tag{27}$$

$$\Phi_{\mathrm{Kr}}(k\ll\lambda_B^{-1},\varepsilon,\chi) = 4\pi^2 k\chi q\sqrt{\nu/\varepsilon}. \tag{28}$$

where the Batchelor length scale $\lambda_B = (\nu D_T^2/\varepsilon)^{1/4}$ and $q$ is a constant taken to be 5.26. This expression does not include a $k^{+1/3}$ inertial–convective subrange, which we ignore here as it increases the integral of the temperature gradient spectrum from $k=0$ to $k=\infty$ by less than 1% and therefore has negligible effect on our results.

A fit against Eq. (28) can be rearranged to give $\chi_{\mathrm{init}}$, which is related to $\chi$ through the correction
function $F_{\mathrm{Kr}}$ as

$$\chi = \chi_{\mathrm{init}}/F_{\mathrm{Kr}}. \tag{29}$$

$F_{\mathrm{Kr}}$ is not raised to a power like $F_{\mathrm{Na}}$ (Eq. (17)). For small values of $k$, $\Phi_{\mathrm{Kr}}\propto\chi$ whereas $\Phi_{\mathrm{Na}}\propto\varepsilon^{2/3}$.

The derivation of $F_{\mathrm{Kr}}$ is equivalent to $F_{\mathrm{Na}}$. We therefore present only the result:

$$\frac{1}{k_\mathrm{h}-k_\mathrm{l}}\int_{k_\mathrm{l}}^{k_\mathrm{h}}\frac{H_{Tt}^2(k)\,\Phi_{\mathrm{Kr}}(k,\varepsilon,\chi_{\mathrm{init}}/F_{\mathrm{Kr}})}{4\pi^2 k(\chi_{\mathrm{init}}/F_{\mathrm{Kr}})q\sqrt{\nu/\varepsilon}} - F_{\mathrm{Kr}} = 0. \tag{30}$$

Note that $F_{\mathrm{Kr}}(\varepsilon,\chi_{\mathrm{init}},W)$ depends on the underestimate $\chi_{\mathrm{init}}$, but the 'true' or 'corrected' value of $\varepsilon$ calculated in Sect. 4.





### 6.2 Obtaining $\chi_{\text{init}}$ from fast thermistor voltage spectra

Like we did for $\varepsilon_{\text{init}}$ in Sect. 4.2, we derive the expression for $\chi_{\text{init}}$ in three steps. First, equate the right hand sides of Eqs. 11 and 28:

$$\left( \frac{C_{2T} + 2C_{3T} \langle V_T \rangle}{C_{Tt}} \right)^2 \frac{1}{W} \Psi_{Tt}(f) = 4\pi^2 k \chi q \sqrt{\nu/\varepsilon}. \tag{31}$$

Then, rearrange while substituting $k = f/W$ to get

$$\chi_{\text{init}} f^1 = \frac{1}{4\pi^2 q \sqrt{\nu/\varepsilon}} \left( \frac{C_{2T} + 2C_{3T} \langle V_T \rangle}{C_{Tt}} \right)^2 \Psi_{Tt}. \tag{32}$$

Finally, solve for $\chi_{\text{init}}$ using a least-squares fit (Appendix C):

$$\chi_{\text{init}} = \frac{1}{4\pi^2 q \sqrt{\nu/\varepsilon}} \left( \frac{C_{2T} + 2C_{3T} \langle V_T \rangle}{C_{Tt}} \right)^2 \underbrace{\frac{\sum_f f \Psi_{Tt}}{\sum_f f^2}}_{\text{Calculate onboard}}. \tag{33}$$

### 320 6.3 Quality control of the temperature gradient spectral fits

The approach to quality controlling the fast thermistor data is the same as that for shear (Sect. 4.3). That is, we fit $\Psi_{Tt}$ over $f_l$–$f_m$ and $f_m$–$f_h$ (1–3 and 3–5 Hz). This ultimately provides two estimates of $\chi$ for each spectrum, which are combined as follows:

$$\chi = \text{mean} \left( \chi_{\text{l–m}}, \chi_{\text{m–h}} \right) \tag{34}$$

$$\chi \text{ fit score} = \frac{\min \left( \chi_{\text{l–m}}, \chi_{\text{m–h}} \right)}{\max \left( \chi_{\text{l–m}}, \chi_{\text{m–h}} \right)}. \tag{35}$$

We do not apply a low $\chi$ threshold equivalent to Eq. (26).

### 7 Test of the reduction scheme for $\chi$

Profiles of $\chi$ from the reduced scheme compare well to the standard processing, albeit with a small bias in one direction for low values and in the other direction for high values (Fig. 8). Across all values, the 330 two approaches agree within a factor of two 78% of the time (Fig. 9). By comparison, 82% of segments exhibit a factor-of-two agreement between $\chi$ values from the two fast thermistors on the same unit.



Compared to shear spectra, non-dimensionalized temperature gradient spectra have lower fit scores. Especially for the lowest fit scores, the measured temperature gradient spectra tend to be too high at lower frequencies and vice versa (Fig. 10).




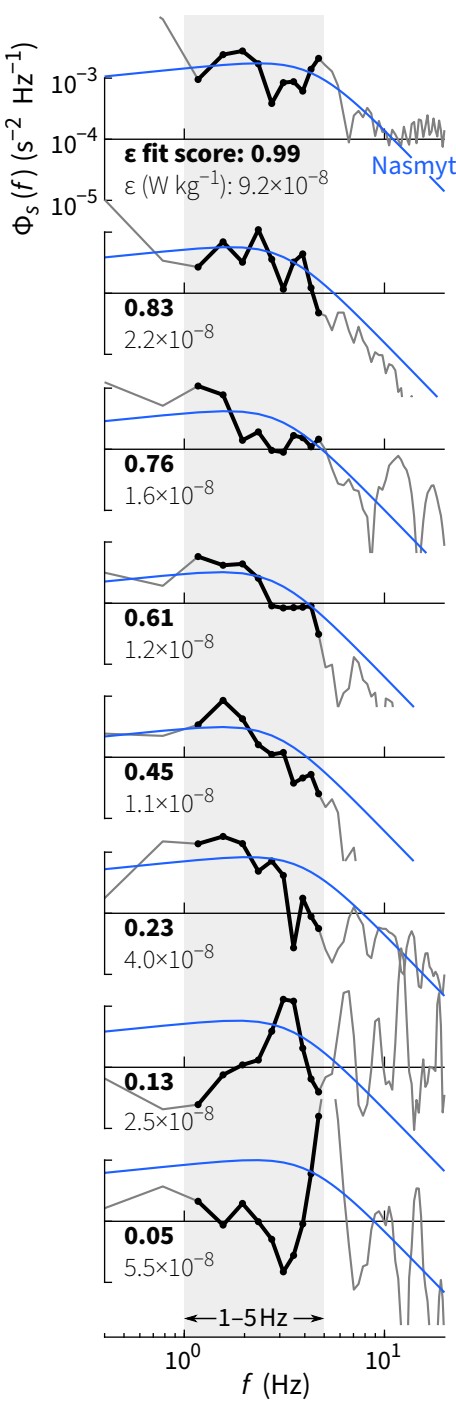

**Figure 4.** Examples of measured shear spectra exhibiting a range of $\varepsilon$ fit scores (Eq. (25)). The best fit is at the top with progressively worse fits (lower scores) moving downward. Each score is only based on spectral coefficients from 1–5 Hz, but lower and higher frequencies are shown for reference.




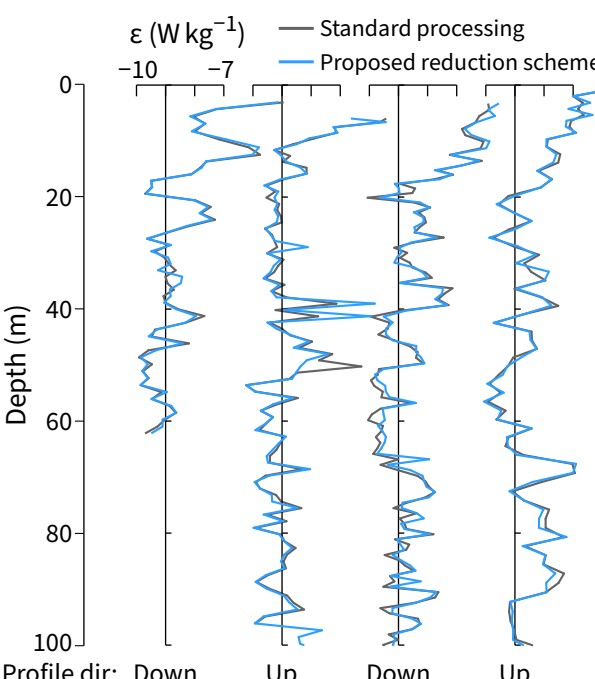

**Figure 5.** Testing the proposed data reduction scheme for shear measurements against the standard processing approach. One upward and one downward profile from each of the two FCS units were arbitrarily chosen for this comparison.



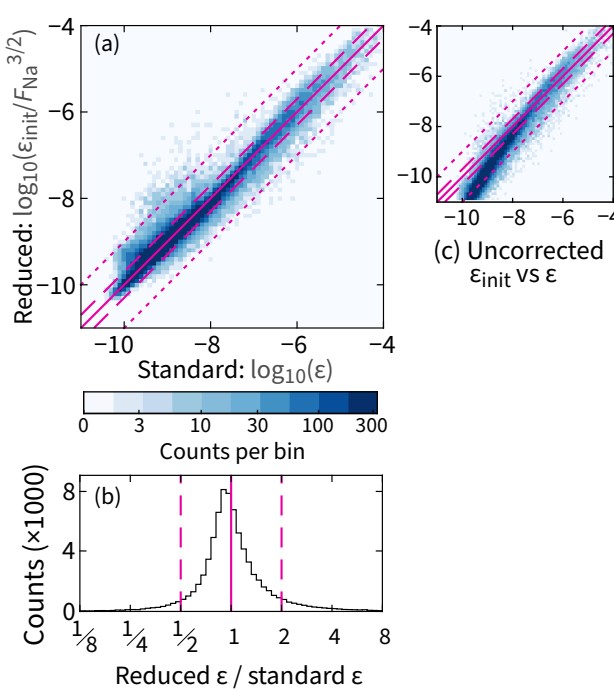

**Figure 6.** Statistical test of the proposed data reduction scheme for $\varepsilon$ based on all 650 profiles (77 000 segments). (a) A comparison that includes the dependence on $\varepsilon$. (b) Further summarized data that exclude this dependence. (c) As for panel a, but uncorrected ($F_{Na} = 1$).

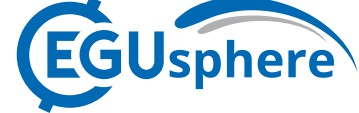

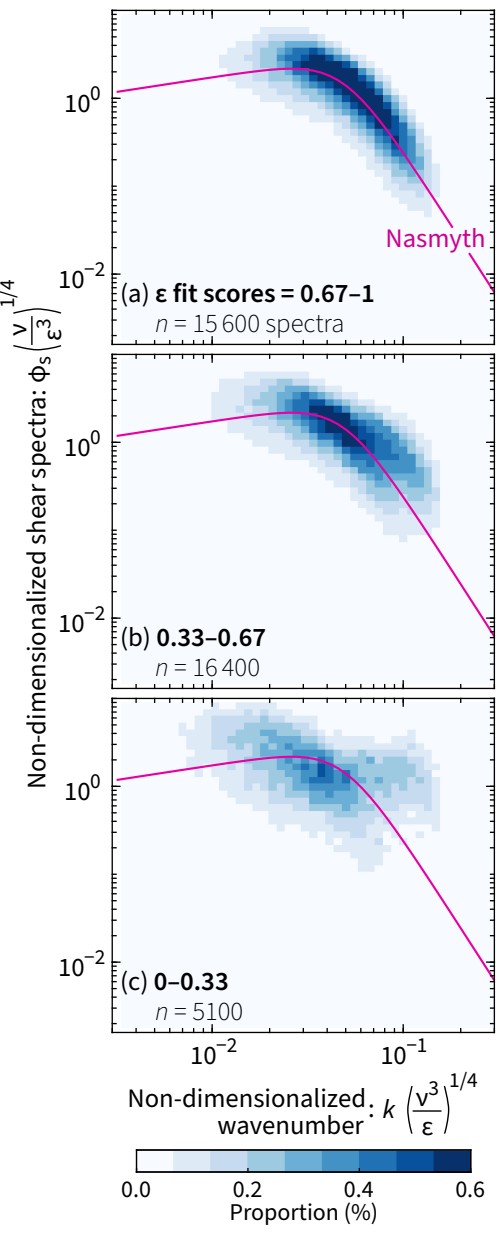

**Figure 7.** As the $\varepsilon$ fit score decreases from top to bottom, there is a corresponding decrease in the level of agreement and tightness of spread between (i) non-dimensionalized, measured shear spectra and (ii) the Nasmyth spectrum. These two-dimensional histograms include only spectral cofficients with frequencies between $f_l$ and $f_h$.





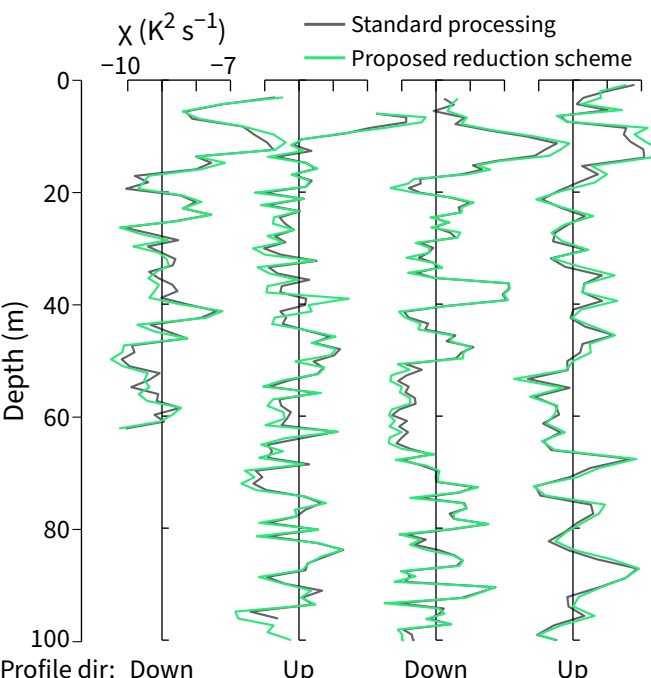

**Figure 8.** Testing the proposed data reduction scheme for fast thermistor measurements. The profiles used are the same as those chosen in Fig. 5.





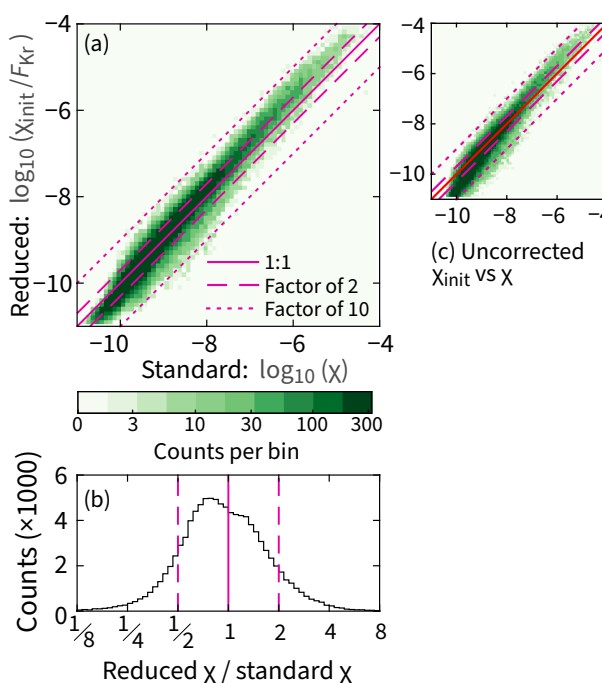

**Figure 9.** Statistical test of the proposed data reduction scheme for $\chi$. Equivalent to Fig. 6 except for $\chi$ not $\varepsilon$. In total, there are 100 000 segments of data.



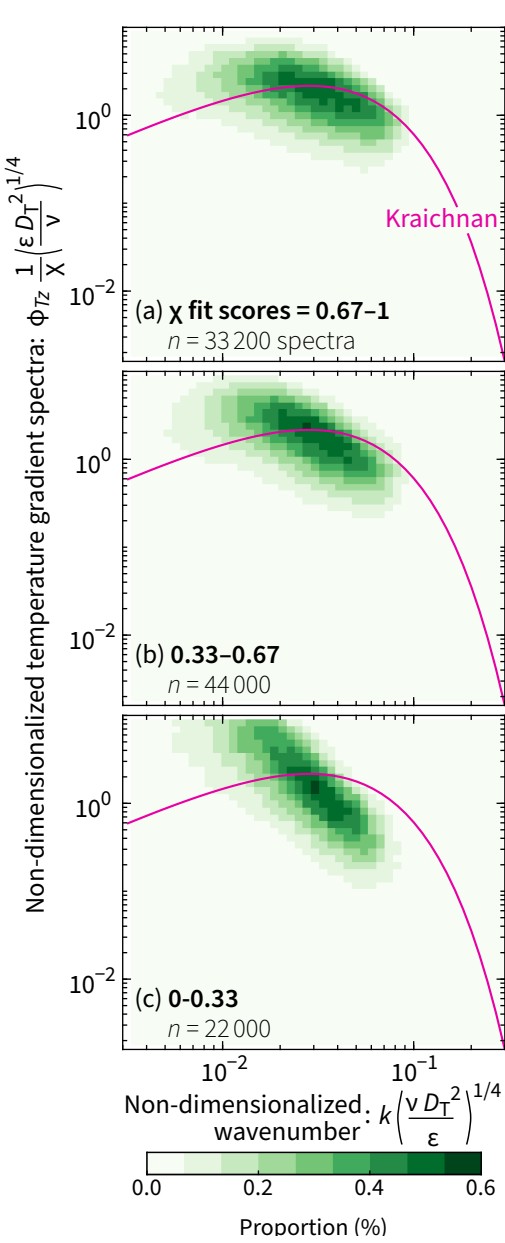

**Figure 10.** As for Fig. 7, but for temperature gradient rather than shear.




## 8 Recommendations

### 8.1 Setting the scheme's parameters

Our scheme requires a few user-defined parameters: $f_l$, $f_h$, $N_{seg}$, and $N_{fft}$. For this paper, we based these partly on the profiling speed and scientific goals of FCS. For a different profiler, we suggest the following:

– Choose $f_h$ based on a typical profiling speed such that $k_h = f_h/W \approx 25\,\mathrm{cpm}$ for a nominal profiling speed $W$. For a wide range of $\varepsilon$ values, 25 cpm is close to, or beyond, the peak of the Nasmyth spectrum (Fig. 1). Further, $\varepsilon$ can be sensitive to the wavenumber fitting range, but centering the fit near $k \approx 10$–20 cpm minimizes this sensitivity (Bluteau et al. 2016). As an example, if FCS profiled at $\sim 0.5\,\mathrm{m\,s^{-1}}$, we would consider setting $f_h \approx 12\,\mathrm{Hz}$.

– Keep $f_l$ in the range 0.5–1 Hz. Although our scheme uses the inertial subrange (i.e., low frequencies/wavenumbers) as its starting point, there is little to be gained by including frequencies of $\mathcal{O}(0.1)\,\mathrm{Hz}$. A possible exception, albeit tangential to this paper, is if the turbulence measurements come from a platform that effectively measures horizontally. In such cases, FFT segments may be many minutes or more and thereby contain useful low-frequency information (e.g., Bluteau et al. 2011; Moum 2015).

– Ensure that there are no known issues such as vibrations that are likely to adversely affect spectral cofficients within the $f_l$–$f_h$ range. Although vibrational effects can be removed spectrally (Goodman et al. 2006), doing so is beyond the scope of our scheme.

– Define $f_l$ and $f_h$ separately for shear and temperature gradient if appropriate. Although we set them equal here, this is not necessary.

– Choose $N_{seg}$ and $N_{fft}$ based on scientific goals and, possibly, any logistical constraints; the data reduction scheme is agnostic to these numbers. For example, at the expense of vertical resolution, we could halve the file size of our transmitted dataset by doubling $N_{seg}$ from 512 to 1024.

– Reasonable choices for $N_{fft}$ are $N_{seg}/2$ or $N_{seg}/4$, which correspond to three or seven half-overlapping subsegments, respectively. There is little to be gained by diving a segment into even more subseg-





ments so as to produce smoother spectra before fitting. As Ruddick et al. (2000) notes, the task is analogous to fitting a line to 20 points at once or first clumping them in groups of, say, five and then fitting the four averaged points.

## 8.2   Evaluating the reduced data

One step that cannot be automated is the heuristic evaluation of the reduced turbulence data after they
have been converted from voltage quantities to physical ones. For this evaluation, we recommend looking into multiple quantities. First consider the fit scores (Eqs. 25 and 35). These scores work well, but they are not a perfect measure of fit. They should be used together with other quality control checks such as comparing

    – $W$ and $W_{\min}$ (Eqs. 13 and 14) to check whether the profiling speed is constant over a segment;

– $\varepsilon$ values from the two shear probes; and

    – turbulent features in successive profiles.

The last point is most applicable for a vertical profiler cycling rapidly – for example, twice per hour for FCS. In this case, the profiler is nominally sampling the same vertical fragment of the ocean on a time scale comparable to that over which turbulence evolves. In our experience, many turbulent patches extend
over 5–10 profiles.

## 9   Conclusions

We have developed a data reduction scheme applicable to vertical profiling of turbulence variables in which each $\sim$5 s segment is distilled to 15 quantities (Fig. 11). In post-processing, we reconstruct estimates of $\varepsilon$ and $\chi$, associated quality control metrics, and other quantities such as the temperature and
profiling speed. The raw data that go into the 15 quantities are seven different voltages ($V_P$; $V_T$ and $V_{Tt}$ for each thermistor; and $V_s$ for each shear probe). Hence, for each 512-element segment, we effectively reduce the data by a factor of $512 \times 7/15 \approx 240$.

    This reduction compresses the output data file size for each dive from megabytes to kilobytes. For example, the total amount of data per dive (two profiles) can be estimated assuming our nominal dive depth





## PRE-DEFINED PARAMETERS

$N_{\text{seg}} = 512$, $N_{\text{fft}} = 256$, $f_l, f_m, f_h = 1, 3, 5\,\text{Hz}$, $C_{2P} = 77\,\text{psi}\,\text{V}^{-1}$ (Appendix B)

## ONBOARD CALCULATIONS

**Reshape raw voltage signals**
Convert each 1D signal to a 2D array $(N_{\text{blk}}, N_{\text{seg}})$

**Discard non-profiling data**
Use $W_{\text{min}}$ threshold ($\propto \min|\Delta V_P(t_i)|$, Eq. 14, Appendix B)

**Average voltages over the $N_{\text{seg}}$-length blocks**
$\langle V_{T1}\rangle, \langle V_{T1}^2\rangle, \langle V_{T2}\rangle, \langle V_{T2}^2\rangle, \langle V_P\rangle, \Delta V_P$ (Eq. 13)

**Despike shear voltages**
Apply $3\sigma$ threshold to $V_{s1}$ and $V_{s2}$ (Appendix E)

**Calculate voltage spectra**
$\Psi_{s1}(f), \Psi_{s2}(f), \Psi_{Tt1}(f)$, and $\Psi_{Tt2}(f)$

**Fit shear spectra over two ranges**
$$
\left.
\begin{array}{ll}
\sum_f f^{1/3}\Psi_{s1}(f)/\sum_f f^{2/3} & \text{over } f_l{-}f_m \\
\quad\text{—\,"\,—} \qquad \text{—\,"\,—} & \text{over } f_m{-}f_h \\
\sum_f f^{1/3}\Psi_{s2}(f)/\sum_f f^{2/3} & \text{over } f_l{-}f_m \\
\quad\text{—\,"\,—} \qquad \text{—\,"\,—} & \text{over } f_m{-}f_h
\end{array}
\right\} \text{(Eq. 23)}
$$

**Fit $Tt$ spectra over two ranges**
$$
\left.
\begin{array}{ll}
\sum_f f^{1}\Psi_{Tt1}(f)/\sum_f f^{2} & \text{over } f_l{-}f_m \\
\quad\text{—\,"\,—} \qquad \text{—\,"\,—} & \text{over } f_m{-}f_h \\
\sum_f f^{1}\Psi_{Tt2}(f)/\sum_f f^{2} & \text{over } f_l{-}f_m \\
\quad\text{—\,"\,—} \qquad \text{—\,"\,—} & \text{over } f_m{-}f_h
\end{array}
\right\} \text{(Eq. 33)}
$$

## POST-PROCESSING

**Calibrate averaged voltages**
$T_1$ and $T_2$ (Eq. 6), $P$ (Eq. 12), and $W$ (Eq. 13)

**Derive viscosity and thermal diffusivity**
Use measured $T$ and $P$ together with $S$ from SOLO-II CTD

**Calculate four 'initial' turbulent dissipation values**
For $S_1$ and $S_2$, get $\epsilon_{\text{init}}$ for $f_l{-}f_m$ and $f_m{-}f_h$ (Eq. 23)

**Repeat above step for thermal dissipation**
For $Tt_1$ and $Tt_2$, get $\chi_{\text{init}}$ for $f_l{-}f_m$ and $f_m{-}f_h$ (Eq. 33)

**Calculate the correction factors**
$F_{\text{Na}}$ (Eq. 20) and $F_{\text{Kr}}$ (Eq. 30)

**Correct initial estimates**
$\epsilon_{\text{init}} \rightarrow \epsilon$ (Eq. 17) and $\chi_{\text{init}} \rightarrow \chi$ (Eq. 29)

**Combine $f_l{-}f_m$ and $f_m{-}f_h$ fit values**
$\epsilon$ (Eqs. 24 and 26) and $\chi$ (Eq. 34)

**Calculate goodness of fit of spectra**
$\epsilon$ fit score (Eq. 25) and $\chi$ fit score (Eq. 35)

**Figure 11.** Summary of the data reduction scheme. Each 512-element segment of data is ultimately compressed down to the 15 highlighted quantities that are then transmitted. These are calibrated and/or converted into turbulence quantities in post-processing.

and profiling velocity of 120 m and $0.2\,\text{m}\,\text{s}^{-1}$. Each dive creates $15 \times 2 \times 120\,\text{m}\,/\,(0.2\,\text{m}\,\text{s}^{-1} \times 5.12\,\text{s}) \approx$ 3500 quantities. Transmitting each quantity as a 16-bit word equates to approximately 7 kB per dive.

     One luxury we lose is the ability to inspect the raw signals. Typically this would help to (i) cultivate faith in the data, (ii) flag which segments to discard, and (iii) inform work-arounds such as filtering out potential narrowband vibrations in shear spectra. Our scheme accounts for this constraint in two ways.




First, we fit spectra over relatively low frequencies (1–5 Hz) that are unlikely to be affected by noise or vibration. Second, we reduce the data in a way that uses as little arithmetic as possible. Obviously, we cannot reverse-engineer the raw signals, but by making the onboard calculations simple we give ourselves the best chance to later fix or identify any unforeseen issues.

Although the onboard reduction eliminates possibilities in how we process turbulence data, it opens
up possibilities in how we obtain turbulence data. By visualizing how turbulence evolves over successive dives in near-real-time, we can concentrate on regions of interest by adapting the dive schedule to profile more frequently or to different depths. If instead we encounter quiescent periods, we might consider profiling less frequently and thereby conserving battery life. Our ultimate objective is to treat FCS floats as expendable.

**Appendix A: Transfer functions for FCS sensors**

Voltage signals from shear probes and thermistors are a smoothed representation of the true environmental signal. If the smoothing is a spatial effect, it is described by a transfer function $H^2(k)$. If the smoothing is a temporal effect, it is more natural to use $H^2(f)$. We can use these interchangeably because $f = Wk$ and therefore $H^2(f) = H^2(Wk)$. For FCS, there are three components to the transfer function for each sensor:

$$H_s^2(k) = H_{\mathrm{SP}}^2(k)\, H_{\mathrm{AA}}^2(f)\, H_{\mathrm{D}}^2(f) \tag{A1}$$

$$H_{Tt}^2(k) = H_{\mathrm{FT}}^2(f)\, H_{\mathrm{AA}}^2(f)\, H_{\mathrm{D}}^2(f) \tag{A2}$$

where we have used the following shorthand: SP = shear probe, FT = fast thermistor, AA = anti-aliasing, and D = digital. We describe each of these in turn.

Shear probes built and calibrated by the Ocean Mixing Group are very close in dimension to those
examined by Ninnis (1984) who measured their wavenumber response and represented it as

$$H_{\mathrm{SP}}^2(k) = \sum_{n=0}^{4} a_n \left(\frac{k}{k_0}\right)^n \tag{A3}$$

where $a_0 = 1.000, a_1 = -0.164, a_2 = -4.537, a_3 = 5.503, a_4 = -1.804,$ and $k_0 = 170\,\mathrm{cpm}$.





Temporal averaging of temperature at high frequencies due to the thermal response of the fast thermistor is modeled following Sommer et al. (2013) and Lien et al. (2016):

$$H_{\mathrm{FT}}^2(f) = \frac{1}{\left(1 + (2\pi f \tau)^2\right)^2} \tag{A4}$$

where $\tau = 0.01\,\mathrm{s}$ so that $H_{\mathrm{FT}}^2(5\,\mathrm{Hz}) = 0.83$. Note that there is large sensor-to-sensor variation among thermistors, which means there is not one true thermal response correction (Nash et al. 1999). Compared to Sommer et al. (2013), other nominal corrections tend to be less aggressive (see, e.g., Bluteau et al. 2017). Our reduced scheme is built in such a way that a different correction can be applied in post-
processing if desired.

Raw shear and thermistor voltage signals are both subject to two filters. First, an analog antialiasing filter (two-pole Butterworth) with an $f_{\mathrm{c}} = 40\,\mathrm{Hz}$ cut-off:

$$H_{\mathrm{AA}}^2(f) = \frac{1}{1 + (f/f_{\mathrm{c}})^4}. \tag{A5}$$

After the analog signal is anti-aliased, it is digitized at $400\,\mathrm{Hz}$. Before subsampling to the final $100\,\mathrm{Hz}$
output, the signal is digitally filtered. For the 2019 FCS cruise, the signal was convolved with a symmetric 29-element kernel in which the first 15 elements were

$$g_i = (2^{16} - 1)^{-1} \times [52, 221, 393, 427, 174, 0, 0, 0, 0, 0,$$
$$1970, 5054, 8202, 10558, 11433]. \tag{A6}$$

This is a sinc kernel but with negative values set to zero. (We are currently investigating better choices for
future implementations). The filter has a half-power ($-3\,\mathrm{db}$) point at $25\,\mathrm{Hz}$.

## Appendix B: Identifying the start and end of a profile

Early in our processing routine, we partition the raw voltage signals into 512-element segments. In order to discard the segments in which FCS was not profiling, we need robust (yet simple) criteria that demarcate the start and end of a profile. For the start, we search for the first three consecutive segments in which
$W_{\mathrm{min}} > 0.05\,\mathrm{m\,s^{-1}}$. For the end, we swap the inequality.

A drawback of this approach is the appearance of a quantity in physical units ($0.05\,\mathrm{m\,s^{-1}}$). This is the one instance where we hard code a calibration coefficient in the onboard software, rather than apply it in



post-processing. Fortunately, the relevant coefficient can be approximated as constant: $C_{2P} = 76.7 \, \mathrm{psi} \, \mathrm{V}^{-1}$

(barring a redesign of the circuitry or the use of a different brand or model of pressure sensor). For the two

units already built, $C_{2P} = 76.81 \, \mathrm{psi} \, \mathrm{V}^{-1}$ and $76.53 \, \mathrm{psi} \, \mathrm{V}^{-1}$. By comparison, among the four shear probes

on the two units, the calibration coefficients vary by 30%.

At least for the initial implementation of our scheme, we do not include an algorithm to detect the

surface to within centimeters. Doing so would let us work backward to put our uppermost depth bin as

close to the surface as possible. However, we expect that this could be a fragile part of the scheme. Further,

FCS lacks a micro-conductivity sensor, which is likely the sensor best suited for identifying the air–sea

interface (e.g., Ward et al. 2014).

Without surface detection, the depths of the uppermost bins will be realized randomly. In the worst

cases, we would discard the top $\sim 1 \, \mathrm{m}$ ($5 \, \mathrm{s}$ at $\sim 0.2 \, \mathrm{m} \, \mathrm{s}^{-1}$). To alleviate this, we may use half-overlapping

bins near the surface. The exact implementation will be determined later in the development.

## 450 Appendix C: Least-squares fitting of power laws

In this paper, we use power law fits to derive turbulence quantities: $\Psi_s = A_\varepsilon f^{1/3}$ and $\Psi_{Tt} = A_\chi f^1$, where

$A_\varepsilon$ and $A_\chi$ are substitutes for the expressions in Eqs. 22 and 32. With only a single parameter for each fit,

implementing a least-squares fit is easy.

Assume we are fitting the vector $\Psi_i$ to the function $A f_i^n$ where $n$ is either $1/3$ or 1. The sum of squared

residuals is therefore

$$\sum r_i^2 = \sum (\Psi_i - A f_i^n)^2 . \tag{C1}$$

The minimum with respect to $A$ is where the derivative is zero:

$$\frac{\partial}{\partial A} \sum r_i^2 = \sum -2 f_i^n (\Psi_i - A f_i^n) = 0 . \tag{C2}$$

Hence,

$$A = \frac{\sum f_i^n \Psi_i}{\sum f_i^{2n}} . \tag{C3}$$

We had originally intended to find $A$ by following Becherer and Moum (2017), who were fitting $f^{1/3}$

spectra. Their simpler method, $A = \sum (\Psi_i / f_i^n)$, is equivalent to a least-squares fit except that the quantity





minimized is the sum of the squares of the *adjusted* residuals, where adjusted means divided by $f^n$. Differences can be ignored when $n = 1/3$, but not when $n = 1$.

## Appendix D: Standard processing of FCS turbulence measurements

The standard processing of FCS turbulence data differs from the reduced scheme in three ways. First, raw data are despiked differently (Appendix E). Second, the 100 Hz raw voltage signals are calibrated into physical quantities right away. Hence, means and spectra are calculated in physical units, not voltage units. Third, the integration of spectra occurs over a variable wavenumber band, which is found iteratively.

When integrating shear spectra (after correction; Appendix A) to find $\varepsilon$, we follow the approach used for the Chameleon profiler (Moum et al. 1995). A first estimate of $\varepsilon$ is made by integrating over $k = 4$–10 cpm. This value provides a first estimate of the Kolmogorov wavenumber $k_s = (\varepsilon/\nu^3)^{1/4}/2\pi$. (The lower limit for Chameleon is 2 cpm, but we increase this for FCS given its slower profiling speed and hence the possibility of contamination by waves at lower wavenumbers.) The upper integral limit is then set to $0.5k_s$ (with a minimum of 10 cpm and a maximum of 45 cpm). The Nasmyth spectra (Eq. (15)) is integrated over the same wavenumber range. If the measured and Nasmyth integrals are within 1%, then $\varepsilon$ is set equal to the integral of the Nasmyth spectrum over all $k$. Otherwise, $\varepsilon$ and $k_s$ are adjusted iteratively until the two integrals agree.

A similar approach is used for integrating $T_t$ spectra to find $\chi$. The model spectrum is the Kraichnan spectrum (Eq. (27)) and, again, the lower limit of integration is 4 cpm. The upper limit is the Batchelor wavenumber $k_b = (\varepsilon/\nu D_T^2)^{1/4}/2\pi$ (with a maximum defined by $kW = 15$ Hz).

## Appendix E: Identifying and removing noise and spikes in the shear signals

To properly despike the raw output of a shear probe requires several steps. Lueck et al. (2018) describe a process in which the signal is high-passed, then rectified, and then low-passed to derive a measure of the local variance. A value is defined as a spike if it is more than eight times (or similar threshold) above the local variance. Spikes are replaced with an average based on surrounding points. This process is then repeated on the new signal, and so on until no spikes are identified.

In our standard processing of FCS data, we use the Lueck et al. (2018) despiking routine. For our
data reduction scheme, we use an approach that is easier to implement and quicker to compute, albeit
less precise. For each 512-element segment of data, a spike is defined as any data point larger than three
standard deviations from the mean. These spikes are replaced by the mean of remaining values in the
segment.

*Code availability.* Our Matlab implementation of the processing code is available from github.com/OceanMixingGroup/
flippin-chi-solo.

*Data availability.* Raw and processed data for the 2019 experiment are available at doi.org/10.5281/zenodo.5719505
or kghughes.com/data.

*Author contributions.* KGH designed the reduction scheme and led the writing of the paper. All authors contributed
to the final version. JNM and DLR lead the development of the FCS profiler on which much of the paper is based.

*Competing interests.* The authors declare that they have no conflict of interest.

*Acknowledgements.* This work was funded by the Office of Naval Research under grant N00014-17-1-2700 (OSU)
and N00014-17-1-2762 (SIO) and continued as part of the ARCTERX (Island Arc Turbulent Eddy Regional Ex-
change) program under grants N00014-21-1-2878 (OSU), N00014-21-1-2762, and N00014-21-1-2747 (SIO). Engi-
neers who contributed to the design and construction of FCS and its sensors: Craig Van Appledorn, Kerry Latham,
Pavan Vutukur, Mark Borgerson (all from OSU) and Ben Reineman, Kyle Grindley, Jeff Sherman from SIO. Aurélie
Moulin executed initial turbulence processing, and Emily Shroyer provided many helpful comments on early drafts.
Reviews from two anonymous reviewers helped improve many aspects of this paper.





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
