# Peer review of "A turbulence data reduction scheme for autonomous and expendable profiling floats"

_EGUsphere, 2022_

## Community Comment (CC1)

[supplement omitted: unrelated document]

---

## Author Comment (AC1)

We thank all the reviewers for their helpful comments. We have made many revisions to the paper and have detailed our responses to all comments in blue text

Our responses to Anonymous Referee 1's comments begin on page 2

Our responses to Toshiyuki Hibiya's comments begin on page 6

Our responses to Cynthia Bluteau's comments begin on page 8

A version of the manuscript showing the changes from the previous version is appended to the end of this document.

We have made a few other changes that are not directly in response to any of the reviewers' comments.

- The quantity $<V_T^2>$ for each thermistor is no longer part of the transmitted dataset since $<V_T^2> = <V_T>^2$ to a very good approximation, and we already have $<V_T>$. We did not previously recognize this opportunity to further reduce the transmitted dataset.
- Instead of transmitting both $<V_P>$ and $\Delta V_P$ for each segment, we transmit the final $V_P$ value in each segment. The former two quantities can be derived from the final $V_P$ values.
- The two changes above reduced the number of quantities in the transmitted dataset from 15 to 12.
- We have changed the thermal response transfer function (Appendix A) from the one discussed in Sommer et al. (2013) to one shown in Nash et al. (1999). In an earlier review, it was noted that the Sommer et al. correction is overly aggressive. In hindsight, we should have made the change then. We are making it now.
- We added a factor of $H^2_{Tt}$, which was missing from the expression for $\Phi_{Tz}$

**Review by anonymous reviewer**

I think this is an interesting paper describing a good method, but I'm not entirely convinced that there is a great reason for doing the fits on a constant frequency band, besides the convenience. The on-board processing is already complicated enough - ultimately I don't understand why the on-board processing shouldn't be more complete (doing the fits on fixed spatial scales on the wavenumber scaled spectra?) or much simpler (by sending back a more representative voltage spectra and doing the fit on shore)?

Our scheme's on-board component involves just as many steps as the reviewer's "much simpler" approach. That is, the reviewer is suggesting we send back representative voltage spectra (presumably with some band averaging to reduce the file size). That is effectively all we are doing on board. We define two bands (1–3 and 3–5 Hz) and then do a band-averaging of sorts across each one. For shear, for example, we calculate $\sum f^{1/3} \Psi_s / \sum f^{2/3}$ (see Figure 11). This calculation is marginally more complicated than that of a mean.

Taking the "more complete" approach is much more involved as it requires calibrating the shear and temperature gradient data (and the pressure to get profiling speed). And then implementing a nonlinear curve fitting routine. All of these steps introduce possibilities for errors to creep in. Our scheme was developed to circumvent these possibilities.

Getting good estimates when the vertical velocity is not the nominal 0.2 m/s (e.g., near the top of the profiles, as the float comes to the surface) seems to be a very important aspect for the specific instrument discussed.

Yes, one of our scientific goals with FCS is to quantify turbulence in the near surface. From our 2019 experiments we learned that faster profiling is generally better near the surface; if the instrument moves too slowly, then wave orbitals introduce problems. We now note in section 3.4 that segments that are negatively affected by waves have two tell-tale signs: (i) $W_{min}$ is not close to $<W>$ and (ii) the fit scores are low because wave motion introduces variance at low frequencies. This causes spectra to be redder than expected and therefore they do not conform to the Nasmyth/Kraichnan models.

As far as I understand, the results are only obtained by fitting frequencies between 1 and 5 Hz. That corresponds to only 10 points in the spectrum… Separating this results on doing the fit on 5 points. Going from the full-spectrum (100 Hz for 5 sec. = 500 points) to 10 points is ultimately the core of the data reduction scheme. The paper claims that one can estimate accurate rates of dissipation from this narrow frequency range (without fancy despiking or using acceleration data). For these 10 points (for each channel), the fitting method returns 2 fitted values (factor of 5). This additional factor of 5 certainly a nice reduction. I also wonder at what precision that data is returned. Choosing a different number representation, or compression could also help here.

We have added a note in the conclusion that for a given dive:

"Transmitting each quantity as a 16-bit float or integer equates to approximately 6 kB per dive. This can be reduced by one-third if the spectral fit metrics are suitably scaled logarithmically and then transmitted as 8-bit integers."

This is our approach for future FCS deployments.

That being said, the paper is generally clear and the method is well documented. Particularly if the profiling (or horizontal velocity) varies a bit more widely, I would hesitate to really champion that method (because of the relatively narrow and fixed frequency band were the fits are done), but I can see how it might be useful and accurate for the application in question.

In section 8, we describe how our scheme could be adapted to profilers with different nominal velocities. If the chosen frequency band is suitably adapted, there's no reason our scheme cannot apply. Of course, we wouldn't recommend anyone blindly apply our scheme to a different profiler without thoroughly testing with an existing dataset from that profiler.

A few more comments:

Line 23-24: It would be useful to state the size of a typical dataset from an Argo float (one profile every 10 days). Something like "In contrast, a typical Argo profile (once every 10 days) contains XXX kB of data).

One of our goals with FCS is to profile the upper ocean rapidly—to about 120 m once every 30 minutes. Obviously, this is very different to the conventional Argo profile of 2000 m every 10 days. Hence, the comparison to the file size of a typical Argo profile is not relevant here. It is only the rate of transmission that we care about.

We have revised the Introduction to make our scientific goals, and hence our data transmission needs, more explicit.

Line 54: Why is there a 3-axis accelerometer, compass, and pitot tube? What is done with that data? In addition to the data compression, if might be worth it to discuss power consumption…

We use the accelerometer to calculate surface wave height spectra when the profiler is surfaced (though the method is detailed elsewhere).

Otherwise, accelerometer, compass, and pitot data are all recorded internally. We do this in case we can recover the profiler and obtain the full, raw dataset. These auxiliary data are useful but not essential, and so they are not transmitted in any way.

In hindsight, noting that FCS includes a compass and pitot tube does not help the reader in any way, so we have removed any description of these two sensors.

Power consumption is certainly part of our research group's discussion, but this is outside the scope of the paper. More generally, engineering details for FCS are described in a separate paper that currently has a status of Minor Revisions for *J. Atmos. Oceanic Tech.:*

*Flippin' $\chi$SOLO, an Upper Ocean Autonomous Turbulence Profiling Float*
J. N. Moum, D. L. Rudnick, E. L. Shroyer, K. G. Hughes, B. D. Reineman, K. Grindley, J. Sherman, P. Vutukur, C. Van Appledorn , K. Latham, A. J. Moulin and T. M. S. Johnston.

Line 148: Could horizontal velocities impact the estimate of *W*? In particular, wave motion will have some horizontal component that is not present in pressure, but does advect turbulence past the sensor, no? In other words, are there situations where the flow past the sensor is not strictly vertical?

Based on analyses not described in the paper, we have found that—to a first order approximation—FCS as a whole is advected by waves. Therefore, the flow past the sensor relative to the instrument's body is independent of wave velocities, both horizontal and vertical. Of course, this is only an approximation and there could be nonzero residual horizontal velocities. We are assuming these are small such that they would have no effect if added in quadrature to $dp/dt$.

More generally, we are aware that $W = dp/dt$ is not a perfect measure of the speed of the sensors relative to the surrounding fluid. Indeed, the main reason that we include the $<W_{min}>$ quantity in our scheme is to identify segments in which $W$ is questionable.

L170: "two-stage approach". This phrasing, and the following sentence, made me expect that a description of the second stage would immediately follow. As it is now, I'm not sure I can readily identify the second stage (not mentioned until line 188).

We have added a sentence so that the expected parallel structure occurs. The paragraph now reads "Here we develop a new and simpler two-stage approach to fitting shear spectra to $\Phi_{Na}$. In the first stage, …. In the second stage, …."

When the initial fit on the voltage is done on a frequency range past the inertial subrange, it seems that the least-square fit would be really dominated by the lower frequency elements of the band, since the spectrum rolls off so rapidly. The fit then doesn't really help with any noise (for example, in Fig 3b). That is presumably captured in the score.

Yes, the fit value is weighted more by the lower frequencies due to roll off, but

(1) the value of $F_{Na}$ is also weighted more by lower frequencies and so counteracts the effect in question when it comes to calculating $\varepsilon$; and
(2) the roll off between 1 and 3 Hz and 3 and 5 Hz is typically small enough that the effects are limited.

Regarding point 1, consider Figure 1b, which uses one band (1–5 Hz) instead of two (1–3 and 3–5 Hz). The value of $F_{Na} = 0.238$ comes from an integral over 5–25 cpm. Half of the this 0.238 value comes from the first 17% of the 5–25 cpm range.

Regarding point 2, except for $\varepsilon < 10^{-9}$ W/kg, the difference in $\Phi_{Na}$ at 1 Hz and 3 Hz (nominally 5 and 15 cpm) is a factor of 5 or less. And similarly for 3 and 5 Hz. In other words, the reviewer's concern is most relevant for low $\varepsilon$ (which makes sense because that is when the roll off will be largest). However, as we note in section 4.3, issues with low $\varepsilon$ are less concerning to us because they "have minimal effect on any averages given that turbulence distributions have high kurtosis, so high values dominate means".

All the fits in Fig 4 have about the same value of epsilon. It might be interesting to have a column in Fig 4 for much smaller values ($10^{-10}$), and larger ($10^{-6}$, say), to see how the fit is affected by what frequency/wavenumber range it is done over…

We have changed Fig 4 so that it now has three columns with five panels per column, not one column with eight panels. The left column has examples with $\varepsilon = 10^{-9}$–$10^{-8}$ W/kg. The middle column has $\varepsilon = 10^{-8}$–$10^{-7}$ W/kg. The right column has $\varepsilon = 10^{-7}$–$10^{-6}$ W/kg.

Raw data are typically not going to be recovered… What is reason for sampling so fast, if only data up to 5 Hz are used? Naively, perhaps, an analog filter could be used and microstructure signal could be sampled slower, no? Would that save power?

Although our scheme is designed so that FCS is expendable, we may still recover the instruments on occasion, in which case we prefer to have the full 100-Hz raw data available. Further, any power savings are unlikely to be worth the effort of redesigning the existing circuits.

**Review by Toshiyuki Hibiya**

**Summary**

Considering that continuous turbulence observations using autonomous and expendable profiling floats such as Deep Argo floats will be the norm in the near future, the development of the data reduction scheme as described in this paper is indispensable and deserves publication. Nevertheless, I am still not convinced about some aspects of the data reduction scheme described in the paper, so that I would be happy to receive some answers before publication.

Thank you for the review.

**Major Comment**

1. The data reduction scheme proposed in this paper presupposes the existence of a spectral slope with $k^{1/3}$ dependence in the inertial subrange. However, Figure 4 shows that the shape of the measured shear spectrum significantly deviates from that of the Nasmyth spectrum, and does not appear to have the $k^{1/3}$ slope presupposed in the inertial subrange. I am afraid that, in this case, the proposed formulation to obtain $\varepsilon$ using the correction factor $F_{Na}$ defined by (17) and (20) might break down.

Our scheme does not presuppose that measured spectra have $k^{1/3}$ slopes. Yes, a $k^{1/3}$ slope is initially assumed but, as we note in the paper:

"Our inertial subrange assumption is often false. Indeed, 'assumption' is perhaps a misnomer as we do not expect it to be true; we know that viscous roll off will often occur at frequencies lower than 5 Hz (25 cpm for a nominal value of $W$ = 0.2 m/s)."

The statement above is key to understanding our unorthodox approach to calculating $\varepsilon$. However, it is buried somewhat in the middle of section 4.1. Therefore, we have added the following paragraph right at the beginning of section 4 to alert the reader to this point:

"In this section, we are ultimately going to fit measured spectra to an inertial subrange model that does not necessarily apply at the relevant frequencies or wavenumbers. We will elaborate as we go, but we want to emphasize in advance that measured spectra do not need to conform to an inertial subrange model for us to obtain accurate values of $\varepsilon$. The inertial subrange is merely a convenient starting point."

Perhaps the best evidence that $F_{Na}$ and hence $\varepsilon$ is correct is the agreement between $\varepsilon$ from the reduced scheme and that from our standard processing (Figs 5 and 6).

2. Also, in this case, does the "fit score" defined in this study have any meaning? In other words, even if a good fit score is obtained by matching the $\varepsilon$ calculated for 1-3 Hz with that for 3-5 Hz, this cannot necessarily be an indicator of a good match between the measured spectrum and the Nasmyth spectrum, and it may cause errors in the estimation of $\varepsilon$ using (17) and (20), right?

The fit score is simply the ratio of two independent values of $\varepsilon$: one from 1–3 Hz and one from 3–5 Hz. Each of these are derived from their respective values of $\varepsilon_{init}$ and $F_{Na}$. As shown in Figure 3a, the two dark blue lines match up, which is equivalent to saying that the two $\varepsilon$ values match. These lines match up because the original measured spectrum (brown line) looks like a Nasmyth spectrum. The opposite is

true for Figures 3b and 3c—the dark blue lines don't match—because the measured spectra don't match well with the Nasmyth spectrum. So, yes, a high fit score does provide an indicator of a good match between the measured spectrum and the Nasmyth spectrum.

3.  In section 4.1, the method for obtaining $\varepsilon_{init}$ is not presented, and the discussion in section 4.1 proceeds without clarifying the definition of $\varepsilon_{init}$. Wouldn't it be easier to understand the overall flow of the discussion if the definition of $\varepsilon_{init}$ written in section 4.2 were given first, followed by the discussion in section 4.1?

The original text did define $\varepsilon_{init}$ but it was an implicit definition and hidden somewhat in the middle of a paragraph. We have reworded a few paragraphs in section 4.1 to highlight the definition of $\varepsilon_{init}$ and point the reader to the fitting method in Appendix C.

4.  Please add to the end of section 7 the reason why the agreement between the obtained $\chi$ values and those obtained from the standard scheme becomes worse than in the case of $\varepsilon$, even though the method for obtaining $\chi$ from the reduced scheme is basically the same as in the case of $\varepsilon$.

We have added the following paragraph:

"There are three reasons for the poorer fits to temperature gradient spectra compared to that for shear. First, shapes of temperature gradient spectra are often more variable; the best choice for non-dimensional spectral model can be debated (e.g., Sanchez et al. 2011). Second, the temperature gradient fits depend on $\varepsilon$. Uncertainties in $\varepsilon$ propagate into the calculation of $\chi$. Third, for our 2019 experiment, the recorded temperature gradient signals"

Minor comments
5.  Line 198: six times smaller than → eight times smaller than

Good catch. Changed as suggested.

6.  Although $H_s(k)$ is defined in (3), it appears somewhat suddenly in (19) in section 4.1 without any connection to the preceding discussion, which seems a bit awkward.

We have re-ordered the text to get rid of the awkwardness. Previously, the two forms of $F_{Na}$ were defined in Eqs 18 and 20, with the awkward discussion of $H_s^2$ in between. Now, the two forms are given in Eqs 18 and 19 and $H_s^2$ is discussed in its own paragraph afterwards.

7.  Just below color tones in Figures 7 and 10: proportion (%) → proportion (× 100 %)

The label is correct as is. There are many hundreds of nonzero bins in these 2D histograms and the upper limit of the colour axes are <1%.

**Review by Cynthia Bluteau**

**Summary**
I initially refused the review request but can now provide comments. I've elected to do so as a community member rather than an anonymous reviewer. My review will be more narrated than usual, given the open peer discussion at EGU journals. My understanding is the authors developed a "reduced algorithm" for estimating two turbulence quantities ($\varepsilon$ and $\chi$) from the voltage spectra of shear and fast-temperature sensors onboard an expendable profiler (SOLO). It needs to be clarified from the methods' description, but the $\chi$ estimates rely on first obtaining $\varepsilon$ from the shear probe. The algorithms are designed to minimize the data transfer rate by fitting a narrow range of frequencies of voltage spectra with a power relationship. This model spectra may not apply over the fitted range (as noted by the authors). Whether the observations are expected to have a power relationship over the fitted frequencies is accounted for with a correction factor $F_{Na}$ (shear) or $F_{Kr}$ (temperature-gradient). They have chosen the empirical Nasmyth (inertial subrange) and the Kraichnan models (viscous-convective subrange) instead of using an inertial model for both datasets or a viscous model for both. Below I summarized further my understanding of the algorithm before providing more details about modifications that may render the article more transparent for readers.

**Algorithm summary**
For each segment of data (e.g., ~5 s chunk of the full profile), the algorithm estimates:

- an average and minimum drop-speed onboard the instrument
- voltage frequency spectra from each temperature and shear probe sensor (are there 2x of each onboard?)
  - Their spectra have roughly 6-10 degrees of freedom in their setup by using 3 overlapping segments of 256 samples each. The spectral bandwidth is from ~0.4 Hz to 50 Hz with a frequency resolution of 0.4 Hz. The drop speed is about 0.2m/s, so their spectra cover wavenumbers ranging from 2 to 250 cpm. Given the thermal frequency response and the spatial size of the shear probe, the spectra are probably "usable" up to 20 Hz (thermistor) and 100 cpm for the shear probe. Of course, noise can limit this further. Still, the drop speed of 0.2m/s nicely "optimizes" the usable range in both the shear and temperature gradient spectra.
  - The calibration coefficients are not stored on the SOLO, unlike recoverable turbulence profilers or most ocean sensors. The lack of calibration constants prevents them from converting the voltage spectra into physical units onboard the SOLO (or onshore since no spectral observations are transmitted).
- Two power fits are performed for each voltage spectrum over a narrow bandwidth of available spectral observations. This bandwidth is between 1 and 5 Hz (1/2 decade of data). The first power fit is between 1 and 3 Hz, and the second is from 3 to 5 Hz. Overall, their 1/2 decade has 10 spectral samples, and each power fit is done with 5 samples.
- Two quantities, one for each power fit, are returned to shore for each voltage spectrum. These quantities are then converted into initial $\varepsilon_{init}$ and $\chi_{init}$ estimates via the calibration constants of the sensors.

- From these initial estimates, the correction factors $F_{Na}$ and $F_{Kr}$ are used to obtain ε and χ. I presume that ε is also fed into $F_{Kr}$ to obtain χ (not clear from the methods' description). Both correction factors depend on the frequencies fitted, the choice of model, etc. It's unclear whether $F_{Na}$ and $F_{Kr}$ are sufficiently general such that any frequencies could be used for fitting the voltage spectra (e.g., different profiling speed or source of vibrations).
- There are accelerometers onboard the SOLO, but they are not used to correct the turbulence (voltage) spectra. Instead, the analysis utilizes data between 1 to 5Hz to avoid surface waves and motion-contamination. The authors state the accelerometers are for computing wave statistics, which I presume will involve additional spectral computations onboard the SOLO. How the turbulence analysis changes in wavy flows dominated by surface waves should be discussed in the ms.

Thank you for providing these summaries of the paper and algorithm. A few clarifications/notes (but mostly leaving our detailed replies to the comments below):

- Yes, there are two shear sensors and two fast thermistors as noted in the second paragraph of section 2.
- We agree that the spectra are "probably useable" up to 20 Hz and 100 cpm, and less if noise is an issue. The operative word here is "probably". Our scheme is necessarily conservative about the upper limits we use for fitting. We need to be confident that the spectra that we are fitting are *useable*, not just *probably useable.* We do not have the luxury of re-examining the raw data or full spectra after they are reduced.
- Our choice to not apply calibration coefficients onboard the instrument is a feature, not a bug, if you will. We do not dwell much on this point in the paper, but postponing calibration has a big benefit: it removes the possibility of an inadvertent mistake (say, the wrong shear probe is installed, or the wrong header is applied) that would cascade through the turbulence profiling algorithm and lead to values of ε or χ that are difficult or impossible to later correct for.
- Yes, values of ε are fed into the calculation of $F_{Kr}$. We have added a sentence to the first paragraph of section 6 to make this clearer.
- The comment about whether $F_{Na}$ and $F_{Kr}$ are sufficiently general is already addressed in section 8.1 in which we provide recommendations for porting our scheme to a different profiler. In short, our scheme is suitable for different profiling speeds but it does assume that any vibrations that occur are outside the fixed frequency fitting range.
- The wave statistics do indeed involve additional spectral computations. On the advice from an early reviewer, these tangential details are left out of the paper.
- See our response to comments 7 and 8 regarding accelerometers and surface wave frequencies.

**Major comments**

I have grouped my concerns into three themes. The main suggestions for implementation in the ms are numbered and shown in italic purple.

1. More transparency is required in discussing the drawbacks of their chosen strategy. The algorithm was designed to limit data transmission at all costs by relying on a small subset of data recorded by the profiler.
2. Lack of assessment of the fit-score as an alternative measure of data quality.

3. Lack of transparency about the χ methods and the data quality both in applying the reduced algorithm and for estimating χ from standard practices.

**1 Reduced algorithm's "framework"**

**1.1 Improvement over using band-averaged (low-resolution) spectra**

The ms should be more transparent in explaining the drawbacks of their chosen strategy when compared to that employed by Rainville et al. (2017). As noted by the authors on L27-31, the reduced algorithms of Rainville et al. (2017) sends band-averaged (i.e., low-resolution) spectra ashore. In their case, 12 spectral observations are transmitted that cover all available lengths and timescales of turbulence, including the noise (a bit overkill, in my opinion). Specifically, their instrument sends 9 spectral samples between 2 to 100 cpm and 4 samples over the wavenumber range (5 to 25cpm) used in the current ms for fitting. The main advantage of sending spectral observations ashore is being free to apply standard fitting or integrating techniques to estimate the turbulence quantities of interest. More importantly, quality-control criteria, such as the mean absolute deviation listed by the authors on L240 can be calculated. This criterion indicates whether the observations follow the expected forms of turbulence, i.e., spectra aren't drowned by noise, motion contamination, or anisotropy. These issues are usually also nicely spotted by inspecting spectra, but with the proposed reduced algorithm in the ms, this information is lost during transmission.

*1. Since the authors only used limited range of turbulence scales (1 to 5Hz), it would be worth highlighting the data "savings" that they gain by transmitting two power fit estimates (1-3Hz, and 3 to 5Hz) over sending for example 4x (band-averaged) spectral observations.*

We have added details to the second paragraph of the Introduction about why Rainville et al.'s scheme is too data intensive for our purposes. In short, with two shear probes, two thermistors, and 12-element spectra, we have 48 values per segment (plus a few other quantities) that must be sent back to shore. For our profiling scenario, we'd end up spending as much time at the surface transmitting data as actually measuring the ocean. This goes against one of our scientific goals of profiling as frequently as possible.

We developed our scheme with an aim of having the smallest possible file size for transmission. That meant we used only two fitting bands (1–3 and 3–5 Hz). But there is nothing to stop someone from using our scheme with three or more fitting bands (say, adding a 5–7 Hz band) and benefiting from any improvements that this entails. We have added this recommendation to our list in section 8.1. We also note in this section that there is a point at which if the number of bands to be used is many more than two, then (as the reviewer alludes) one might as well use the Rainville et al. (2017) instead.

*2. It would also be useful to say why a data reduction scheme that reduces 512 samples to 12 samples (band-averaged spectra) is inadequate, which ultimately resulted in developing an algorithm that fits a narrow range of information onboard the processor.*

As in our reply above, using 12 samples per spectra leads to file sizes sufficiently large that they interfere with and limit our intended profiling strategy.

**1.2 Using a limited range of frequencies for fitting**

The algorithm appears highly dependent and applicable for their particular drop speed, sampling rates and frequencies used for fitting. It also depends on there being no motion contamination over the range

of frequencies used. It makes the paper highly specific for their platform, as opposed to being a "reduced algorithm" for turbulence profilers. This is fine but worth highlighting. However, the reliance on a small subset of the available turbulence information is problematic. Makes you question why even sample the shear signal at rates above 32-64Hz if we can get away with deriving ε (or χ) from such a narrow range of turbulence length/time scales.

We disagree that the scheme is "highly specific" to our platform. Section 8.1 outlines how it can be adapted for other scenarios. Nevertheless, we agree that we had not highlighted enough the assumption of no motion contamination. We now state in the abstract that "..., we focus on a fixed frequency band that we know to be unaffected by vibrations and that approximately corresponds to a wavenumber band of 5–25 cpm."

The reviewer is correct that ε and χ from our reduced scheme would barely change if we had a sampling frequency of 32 or 64 Hz instead of 100 Hz. At the same time, we don't claim that our values of ε and χ are as exact as they could be. Nor are we suggesting that anyone should use our reduced method if they have the luxury of having raw data at 100 Hz or more. But it shouldn't be a surprise that we can get good results using a narrow frequency band. Although a wider band allows for better statistics (as the reviewer notes elsewhere in this review), this doesn't change the fact that shear spectra scale monotonically with ε. And the same for temperature gradient spectra vs χ (provided ε is known).

As a contrived example, consider a scenario in which measured spectra conform perfectly to the Nasmyth spectrum. In this case, one could calculate ε with just one coefficient from the measured spectra (in other words, a very narrow band)

Furthermore, the narrow band of scales used comprises a low number of spectral observations (5x per fit), which have low statistical significance given how the voltage spectra are calculated. The spectra use 3x FFTs with a 50% overlap, each having 256 samples, which is less than typical when accelerometers are used to decontaminate the spectra. Depending on the widowing function applied, this is about 6 to 10 degrees of freedom – a tad more than spectra with no statistical significance. The degrees of freedom wouldn't allow using squared-coherency to decontaminate the spectra (a minimum of 7 NFFTs would be more appropriate). There's all this effort to reduce the data transfer, but an easy saving would be using a statistically significant spectral averaging strategy, e.g., 768 samples for each segment instead of 512 samples (the 5x NFFTs would still have the exact resolution).

Turbulence processing always comes with trade-offs as the reviewer alludes. Yes, we could increase the sampling period and use more subsegments to increase statistical significance, but we then lose vertical resolution. For us, going from 512 to 768 samples equates approximately to $\Delta z$ from 1.0 m to 1.5 m. These may not sound like much, but it matters a lot in places like the near-surface ocean where ε depends strongly on $z$.

Section 8.1 explicitly notes that $N_{seg}$ = 512 and $N_{fft}$ = 256 is merely the choice we made for our own scientific goals. Someone else could easily use our scheme with, say, $N_{seg}$ = 1024 and $N_{fft}$ = 256 to have 7 half-overlapping subsegments per segment.

*3. Would the results be similar if a more statistically significant spectra were used in the calculation? Would band-averaging the spectra change the estimated $\varepsilon_{init}$ ($\chi_{init}$)?*

In the paper with $N_{seg}$ = 512 and $N_{fft}$ = 256, we find that the reduced scheme and standard processing agree within a factor of 2 in 87% of cases for ε and 78% of cases for χ. These became 88% and 78% when we reran the analysis with $N_{seg}$ = 1024 (i.e., 7 subsegments rather than 3). In other words, effectively no improvement. This indicates that statistical significance of spectra is not a limiting factor on the quality of the final dataset.

*4. Fig 3: Include the kinematic viscosity and confidence intervals for these spectra (see §5.4.8 Emery and Thomson, 2001) for calculating the confidence levels*

If we understand correctly, the reviewer is suggesting that we add confidence intervals to the 10 spectral values in each panel. We have not done this as we think it could cause confusion. Figure 3 demonstrates visually how to calculate the ε fit score, which is a value that is independent of confidence intervals. If we include confidence intervals, the reader might incorrectly infer that confidence intervals somehow enter the fit score calculation.

Kinematic viscosity is now noted in the caption.

**1.3 Number of samples used for each fit**

Recent work has concluded that for data with low variance, 8 samples are required for regressions (Jenkins and Quintana-Ascencio, 2020). For highly variable datasets, this number increases to 25. The authors have used 5x spectral observations, compounding the above issue of relying on low bandwidth of turbulence spectral observations that each have low statistical significance. It makes you wonder if the least-square power fit is just a guesstimate of what the turbulence level in the signal might be.

The Jenkins and Quintana-Ascencio results have been quoted without context and they are not relevant here. The numbers 8 and 25 are for their specific and contrived scenario of distinguishing between null, linear, and quadratic data. We are doing something quite different: we have a known model to fit against and we are finding the fit coefficient. We are not trying to distinguish between models.

*5. How large are the confidence intervals for the fitted quantity in Eq 23? Least-square fitting usually allows for this result to be calculated, but these have not been presented.*

As we note in section 4.3, we cannot calculate conventional goodness of fit metrics (including confidence intervals for least-squares fits) with our reduced scheme because we do not know the scaling for each model spectrum until we calculate ε values in the post-processing stage. By this stage, we have lost information about the spectral shape through the summing operation in Eq. (24).

*6. What would be the confidence level for $ε_{init}$ if the spectral confidence levels were propagated, along with the errors associated with using only 5x samples for a least-square fit? The error would propagate through to $F_{Na}$ in Fig 2.*

As per our response to comment 3, we find that statistical significance of the measured spectra is not a limiting factor. And per our response to comment 5, we cannot calculate confidence intervals because we don't know ε until post-processing.

We don't mean to downplay the value of calculating uncertainties for each spectral fit, and we recognize that the reviewer has done a lot of work to help the turbulence community follow best practices on this topic. However, with a tool like FCS, we gain the luxury of a dataset with ~50 profiles per day. Hence, we

personally are better served by using bootstrapping to calculate uncertainties from repeated samples of ε and χ than we are diagnosing the confidence intervals for each individual fit.

**1.4 No reduction of vibration and wave-contamination**
The ms mentioned using accelerometers to determine the wave climate, so spectra are presumably being calculated. It seems no information gained from the accelerometers will be used to assess the quality of the turbulence measurements? Rather than decontaminate the shear spectra, the fit is restricted to frequencies 1 to 5Hz when some platforms do have contamination across those ranges (Fig 5 of Bluteau et al., 2016). Surface waves "seas" are awfully close to the frequency range used here.

*7. The authors should acknowledge that their chosen frequencies avoids motion-contamination for their specific platform.*

As noted earlier, we now state up front in the abstract that the frequency band we use is one that we know to be unaffected by vibration.

*8. It would help to detail in the ms why it's unfeasible to perform this calculation onboard the processor. A squared-coherency estimate (Zhang and Moum, 2010) is mostly rearranging cross-spectral terms (conjugates of FFTs). It is not restricted to being done in physical units. This additional processing could indeed be accommodated by using longer segments. We often live with ADCP profiles with 10 m bins, so getting a lower vertical resolution turbulence signal might be worthwhile if the estimates are more robust.*

On the low frequency side, decontamination via squared coherency is not going to help. The motions that we'd being trying to account for would be surface gravity waves, but the nonlinear relationship between shear and profiling speed makes this too challenging. We now note this in section 3.3. In that same section, we also now note that segments that are adversely affected by waves can be identified in post processing because they typically share two traits: $<W>$ is not close to $W_{min}$ and ε fit score << 1. We state that these segments should be discarded.

For $f$ > 5 Hz, we agree that vibration decontamination is possible and that it can be done in voltage units. But, as the reviewer noted earlier, this requires ≥7 subsegments. To us, this represents too large of a constraint on the vertical resolution. We have added an explicit statement to the end of section 3.1 that we are not pursuing the possibility of motion decontamination.

**2 Fit-score as a quality-control indicator**
The authors' "fit-score" is the ratio of the result obtained using the first 5 samples (1 to 3 Hz) vs the subsequent 5 samples (3 to 5Hz). This quantity tests crudely the sensitivity of the results to a very slight change in frequencies. Having used it myself as a qualitative guide (see Fig 4 of Bluteau et al., 2011), I'm concerned that there's no measure of whether the entire spectra are "garbage" particularly the temperature gradient spectra that are usually much more variable in quality.

If the turbulence shape varies widely over such small changes in frequencies, then, indeed, the data is very poor. The question, though, becomes, how poor? How much variations in shape can we expect? Does this variation depend on the spectral averaging, i.e., the statistical significance of the spectra? Will the fit-score depend on the number of samples used in the individual fits? How do all of these factors translate into rejection criteria? It needs to be clarified if a rejection threshold was proposed. What's

evident in the manuscript is that the fit-score it's now the primary way to assess quality, given that the reduced algorithm does not send spectra observations to shore.

*9. To develop and assess the algorithm, the SOLO was recovered. Thus, it's possible to estimate the mean absolute deviation listed on L240. The mad has existed for 22 years and is on its way to being recommended by the SCOR working group #160 as a quality-control indicator for data archiving. The fit-score should be compared to the mean absolute deviation (mad) for all segments in a scatter plot. The scatter plot would enable readers to judge the robustness and usefulness of the fit-score as a quality-control indicator. Let the data speak for itself.*

Although our fit score and the mean absolute deviation are both goodness of fit measures, they quantify different aspects and there is not an apples-to-apples comparison between the two.

- Mean absolute deviation, as its name implies, focuses on the size of the residuals.
- Our fit score is more focused on whether the residuals are random or autocorrelated.

We have added three sentences to section 4.3 explaining that our fit score focuses on random vs autocorrelated residuals.

To elaborate, consider the spectra examples in Figure 4. Those with the higher scores tend to have unbiased residuals: measured spectral coefficients are just as likely to be above the fit as below it. Those with lower scores tend to have autocorrelated residuals (often manifesting as the sign of the residuals being a function of frequency). Take, for example, a spectrum from Figure 4 whose fit score is low because all coefficients with $f < 3.3$ Hz have negative residuals and vice versa:

[Figure]

This example would also have a large MAD because it's a poor fit full stop. But there are cases where the MAD is large ($\rightarrow$ bad fit) but the fit scores are close to 1 ($\rightarrow$ good fit) because the residuals are large but not autocorrelated.

Obviously, we prefer that our fits don't have large residuals, but the more pressing concern is that the fits do not have autocorrelated residuals.

For what it is worth, we did do the comparison of **

(1) $\varepsilon$ fit score vs MAD($\Phi_s/\Phi_{Na} - \langle\Phi_s/\Phi_{Na}\rangle$) over 1–5 Hz
(2) $\chi$ fit score vs MAD($\Phi_{Tz}/\Phi_{Kr} - \langle\Phi_{Tz}/\Phi_{Kr}\rangle$) over 1–5 Hz

The two quantities are clearly correlated (see figure below), but there is variability for the reasons described above.

[Figure]

**For each fit, we actually calculated the MAD for $\Phi_s/\Phi_{Na}$ and its inverse $\Phi_s/\Phi_{Na}$ and took the larger MAD of the two (and similarly for $\Phi_{Tz}/\Phi_{Kr}$). We did this because there are times when $\Phi_s/\Phi_{Na}$ is small because $\Phi_{Na}$ is large, not because the residuals are small. As a contrived example, $MAD(\Phi_s/\Phi_{Na} - \langle\Phi_s/\Phi_{Na}\rangle) = 0$ if $\Phi_s$ is 0 for every frequency.

**3 Data quality of the temperature gradient data**

**3.1 Methods description**

In general, the χ description is unclear. The ms is organized as if the χ estimates are done in isolation of ε, when Eq 30-31 shows that ε is required to estimate χ. This strategy differs from the many previous chipod papers (e.g., Moum and Nash, 2009; Becherer and Moum, 2017), which use the fast-temperature data to solve for both ε and χ by equating Osborn to Osborn-Cox's model. The proposed algorithm design in the ms was justified by referring to chipod papers pioneered by their research group (Becherer and Moum, 2017). However, the authors haven't highlighted that the new algorithm depends on getting ε first from the shear probe before obtaining χ from the temperature gradient spectra rather than obtaining ε and χ simultaneously from the temperature gradient spectra. Needing both shear probes and fast-temperature sensors in itself increases the amount of data processing and data transmission from the SOLO. Why not get rid of the shear probes completely? Does the SOLO not measure background temperature and salinity?

Inferring values of ε from χ is only possible where there is stratification. Since FCS is an instrument focused on the upper ocean, it will spend a lot of its time in the mixed layer. To measure ε here, it needs shear probes. We now mention this need for shear probes near the end of section 2.

*10. Please explain in the intro [L35-40] whether they are using the shear probe to derive ε, and if this quantity is then used for estimating χ. Some comments as to why this strategy was chosen should be provided given it increases the demands on data processing and data transfer.*

We now state in the last paragraph of the Introduction that, yes, we get ε from the shear probes and subsequently invoke this in the calculation of χ. (This is also reiterated at the start of section 6.) As the reviewer rightly notes, this is different from the chipod method. Consequently, we caused confusion by stating that we were closely following the chipod paper of Becherer and Moum (2017). We have moved and reworded the reference to Becherer and Moum so that it no longer causes confusion.

*11. The ms could also better discuss the implications of relying on ε from the reduced algorithm on the quality of the χ estimates. Unless of course, they've estimated ε and χ simultaneously from the temperature gradient spectra in which case the ms should illustrate how $ε_χ$ compares to ε obtained from the shear probes.*

We have added the following paragraph to section 7 (Testing the reduction scheme for χ)

"There are three reasons for the poorer fits to temperature gradient spectra compared to that for shear. First, shapes of temperature gradient spectra are often more variable; the best choice for non-dimensional spectral model can be debated (e.g., Sanchez et al. 2011). Second, the temperature gradient fits depend on ε. Uncertainties in ε propagate into the calculation of χ. Third, for our 2019 experiment, the recorded temperature gradient signals"

We briefly reiterate this point in section 8.2

"Recall, also, that all uncertainty in ε propagates into the calculation of χ (Sect. 7). If ε for a given segment cannot be trusted, neither can χ."

*12. Also re-iterate the dependency of χ on ε when presenting Eq 30-31, and on L306 when claiming the temperature gradient spectra $Φ_{Kr}$ depends only on χ, which isn't true (see Fig r1).*

The text already includes a reminder of this dependence after Equation 30.

We assume the reviewer meant line 307? This line states that $Φ_{Kr} ∝ χ$, but it does not state it depends *only* on χ. The additional dependence on ε is already clearly stated in the definition of $Φ_{Kr}$ in Eqs 27–28.

**3.2 Information about the chi data quality**

The fact that no temperature gradient spectra were shown in the ms is disconcerting. A few shear probe examples focus on observations between 1 and 5 Hz, which mask any spectral contributions from waves or motion contamination. The only spectral information for χ is the data density plots in Fig 10, which only show observations between 1 and 5Hz. Still, this figure gives an inkling of the temperature gradient data quality. Fig 10b shows a cloud of data with the wrong slope sign at non-dimensional $k ≲ 3 × 10^{-2}$, which then starts to fall off too early. Is this because of problems with the estimated ε shifting the spectra to the left? Fig 10c looks like spectra drowned by noise (perhaps high ε and low χ). Even the high-score examples in Fig 10a could be better. The "peak" data density doesn't fit the Kraichnan form. There's no curvature in the location of maximum data density, unlike Fig 7a for the shear probe.

I'm questioning whether it's the algorithm behaving poorly or whether the collected temperature gradient data could be of better quality at the outset.

*13. To alleviate concerns about the data quality, I strongly recommend adding an extra column in Fig 7 and 10 showing the data density for all wavenumbers, not just those used by the algorithm.*

As suggested, we have added the extra columns to Fig 7 and 10. The new column in Fig 7 shows that, provided the fit score $≳ 0.33$, the measurements agree with the Nasmyth spectrum at all but the highest frequencies where noise becomes an issue. Conversely, the new column in Fig 10 shows looser agreement and that the effects of noise are larger. This implies that the quality of our results is not limited by the algorithm, but by the quality of the temperature gradient data.

*14. Another request is adding an extra column in Fig 4 with the temperature gradient spectra collected concurrently with the shear probe. Preferably all available wavenumbers for data transparency. There are none in the ms, which masks the data quality.*

The purpose of Fig 4 is *not* to demonstrate data quality. It is to demonstrate how well the ε fit score differentiates better and worse fits. We designed the figure to make it easy to follow: the fits get progressively worse moving from top to bottom. Adding a column with the concurrent temperature gradient spectra would confuse the figure as this extra column of fits would not necessarily get progressively worse in the same way.

The temperature gradient spectra certainly are of poorer quality in terms of how well they match the model form. This is now clear from the extra column added to Fig 10 as suggested in the previous comment (and discussed at the end of section 7).

*15. Specify which χ estimates were rejected from the paper. Fig 10 shows variable fit-score, but which would be flagged as unusable for further analysis in a scientific article?*

We now note in section 8.2 that we recommend discarding ε and χ values if their associated fit scores are less than 0.33. As with any turbulence dataset, it is always a challenge to turn a goodness-of-fit continuum into a pass/fail binary. That said, based on our analyses, we find that 0.33 is a good threshold.

*16. Why use the inertial subrange model for shear but then assume the viscous-convective subrange model for temperature when the same frequencies are fitted onboard the processor? Does it not matter that the fitted model isn't expected over the range k?*

There are two subranges for shear (inertial and viscous) and three for temperature gradient (inertial–convective, viscous–convective, and viscous–diffusive). We think the reviewer is implying that we use the viscous subrange for shear so that the two models are more similar? This is not obvious since there isn't a one-to-one correspondence between subranges for shear and temperature gradient.

For our scheme to work, the models used ($\Phi_{Na}$ and $\Phi_{Kr}$) only need to have a subrange in which they are proportional to $k^n$ (or equivalently $f^n$). For shear, we have $k^{1/3}$. For temperature gradient, we have $k^1$. These power law approximations are what makes the onboard component of the scheme simple (see sections 4.2 and 6.2).

To answer the second question in the comment, no it doesn't matter. For elaboration, see our response to the first comment by reviewer Toshiyuki Hibiya.

*17. Also, please use a colour scale/scheme that compares the temperature and shear spectra data density against each other. The colour gradient presents a data count (or proportion of data), but the scheme changes between figures. If a colour theme for shear and temperature is necessary, decorate the x and y-axis colours, but leave the colour gradient the same across all the data density plots (Fig 6, 7 and 8-9).*

We fail to see the reasoning behind using a single colourmap. There is never a need to compare proportions across different figures. Proportions are only meaningful within a single figure.

We use blue for shear and green for temperature gradient throughout the paper to help signal to the reader when the discussion shifts from one quantity to the other.

*18. The colours in Fig 6, 7 and 8-9 also seem to saturate at values below the maximum, which makes it hard to see where the maximum sits relative to the theoretical shape (Fig 7-10) and the 1:1 slope (Fig 6 and 9). Perhaps visit brewermap or cmocean color palettes described in Thyng et al. (2016) article about ocean data visualization.*

We have fixed the colour limits for the affected figures so that they no longer saturate.

Colorbrewer is the tool (via a Matlab wrapper) that we used to generate the blue and green colourmaps used in this paper.

**Other comments**

**Misciting**

Remove the erroneous citations to my published work. On L342, the reference to my article on fitting shear probe data does not state that centring a fit around 10-20 cpm minimizes sensitivity to the fitted range. The only thing that reduces this sensitivity is fitting the correct model (e.g., inertial subrange) over the wavenumbers that this model is expected to be valid. Another way to limit sensitivity is to have high-quality measurements that aren't drowned by noise, vibrations and surface waves. The range of wavenumbers would depend on data quality, drop speed, and model used for fitting. If my results were insensitive to the 10-20 cpm range, I was using a model that covered both the inertial and viscous subranges, and the data was of good quality after decontamination.

We have removed the statement in question.

L344-349. I'd remove the whole paragraph. First, the power fit used by the authors sometimes uses frequencies in the inertial subrange (1 to 5Hz translates to 5 to 25cpm). With the 0.2m/s drop speed, the inertial subrange is only being fitted with the correct model when $\varepsilon > 10^{-7}$ W kg$^{-1}$. For low $\varepsilon < 10^{-7}$ W/kg, the inertial subrange model is fitted to the shear probe's viscous subrange. A similar argument would apply for temperature, except that the authors have chosen to always apply the viscous-convective model instead of the inertial-convective model (Fig. r1).

We have removed the whole paragraph as suggested.

Using a moored platform changes very little other than we have to contend with variable speeds past the sensor. A mooring doesn't automatically translate into long FFT segments. My miscited article (Bluteau et al., 2011) refers to the spectral fitting of the inertial subrange of acoustic-velocity measurements – not shear probes. We use long segments because we're relying on the lower scales of turbulence since the data quality is typically too poor over the viscous subranges (technological issue). Another reason for using long segments is calculating other turbulence quantities such as Re stresses and TKE. These estimates rely on covariances and thus the integration of cospectra with reasonable statistical significance (need more NFFTs, and/or band-averaging).

The reference to Bluteau et al. (2011) was, as we noted, tangential to the paper. We see how it could cause confusion and so have removed it per the previous comment.

[revised manuscript text omitted]

---

## Author Response (AR2)

This version is the same as the previous version with two minor changes.

1. Per the previous file validation comment, Figure 11 is moved prior to the Appendix

2. Per the editor's comment, a reference to Figure 1 has been added to Section 4.1